# TOWARDS EFFICIENT FAIRNESS IMAGE RETRIEVAL WITH DISENTANGLED INFORMATION SUPPRESSION

## ABSTRACT

Deep hashing has emerged as an effective method for large-scale image retrieval, improving computational efficiency by converting high-dimensional data into compact binary codes. Despite its success, recent studies reveal that deep hashing methods may exhibit fairness issues, leading to biased or discriminatory retrieval results across demographic groups. To jointly improve retrieval accuracy and group fairness, we introduce **D**isentangled **I**nformation **S**uppressed **H**ashing (**DISH**), a framework that learns fair and discriminative representations. DISH employs a disentangled encoder to decompose each image into factor-specific representations. To encourage semantic concentration and interpretability, a disentangled consistency objective is introduced to enforce factor-level stability under augmentation and align semantic evidence with latent factors. Furthermore, an information suppression module is designed to mitigate sensitive information leakage through probability-driven channel masking, channel-wise adversarial learning, and conditional covariance regularization. These components work collaboratively to eliminate sensitive signals both within and between feature channels while preserving semantic discriminability. Extensive experiments on multiple benchmarks show that DISH substantially outperforms state-of-the-art deep hashing baselines in retrieval accuracy while achieving better fairness.

## 1 INTRODUCTION

Deep hashing has emerged as an effective approach for large-scale image retrieval tasks. By leveraging deep neural networks, deep hashing methods transform image data into compact binary codes, substantially reducing storage requirements and enabling rapid retrieval through efficient binary operations(Slaney & Casey, 2008; Gong et al., 2012; Liu et al., 2012; Hoe et al., 2021). Compared to traditional retrieval methods, hashing-based approaches offer remarkable advantages in retrieval speed and scalability(Yuan et al., 2020; Wang et al., 2023), making them especially valuable for applications such as search engines(Wang et al., 2012), recommendation systems(Luo et al., 2024).

Despite their success in retrieval accuracy, recent studies have revealed that deep hashing models, similar to many other representation learning systems, may inherit and even amplify societal biases present in training data, leading to systematically unfair retrieval results across demographic groups defined by sensitive attributes such as age, gender or ethnicity(Zhang et al., 2024). Although fairness has been extensively explored in general classification tasks(Berk et al., 2017; Hardt et al., 2016; Nabi & Shpitser, 2018; Zafar et al., 2017; Sattigeri et al., 2019), it remains under-explored in the context of deep hashing: most existing methods focus solely on maximizing retrieval accuracy(Li et al., 2015; 2017; Su et al., 2018; Li et al., 2019). Even when fairness is considered, interventions are often applied post-hoc or confined to the final hash space(Zhang et al., 2024), where the extreme compression and discreteness of binary codes severely limit the capacity to disentangle and suppress sensitive signals without sacrificing semantic utility. Furthermore, directly adapting established fair representation paradigms such as appending a hashing loss to an adversarial fair encoder (e.g., AFR (Zhang et al., 2018)) or disentangled VAE frameworks (Higgins et al., 2017; Kim & Mnih, 2018; Creager et al., 2019) faces inherent challenges. As representations transition from continuous manifolds to the discrete Hamming space, the process risks disrupting the learned manifold structure. This implies that continuous fairness does not automatically guarantee binary fairness, creating a need for a retrieval-specific architecture that enforces fairness robustness.

To tackle these issues, we propose **D**isentangled **I**nformation **S**uppressed **H**ashing (**DISH**), a framework that learns fair and discriminative hash representations by intervening in the continuous feature space prior to binarization. Unlike prior methods that rely on a single global latent vector, DISH constructs a factor structured latent space. DISH employs a disentangled encoder to decompose each image into factor-specific representations, these representations are regularized by a disentangled consistency objective, which promotes semantic concentration and factor-level stability under augmentations. Built upon this structured decomposition, DISH suppresses sensitive cues through a channel-wise adversarial learning and conditional covariance regularization, theoretically minimizing their recoverability both within and across feature dimensions. A final semantic alignment loss in Hamming space ensures the binarized codes retain strong discriminative power, striking an effective balance between fairness and performance.

In summary, our key contributions are as follows:

(1) We propose DISH, a fairness-aware hashing framework that introduces a factor level decomposition to disentangle semantic and sensitive factors in the continuous feature space prior to binarization, enabling targeted suppression of bias while preserving retrieval semantics;

(2) We introduce a theoretically grounded information-suppression mechanism that combines channel-wise adversarial learning with conditional covariance regularization, minimizing sensitive leakage both within and across latent factors with formal guarantees via mutual information bounds;

(3) We conduct comprehensive evaluations on multiple benchmarks, showing that DISH establishes new state-of-the-art results in balancing retrieval performance and fairness.

## 2 RELATED WORK

**Learning to Hash.** Hashing has been widely adopted for large-scale image retrieval due to its computational and storage efficiency. Early work spans data-independent LSH (Slaney & Casey, 2008) and data-dependent schemes such as ITQ and supervised hashing (Gong et al., 2012; Liu et al., 2012; Shen et al., 2015). With the rise of deep learning, hashing has evolved into pairwise, triplet-based, and pointwise paradigms, including methods that preserve pairwise similarity (Wang et al., 2010; Li et al., 2015), enforce triplet ranking constraints (Wang et al., 2017), or directly supervise hash codes via semantic prototypes (Su et al., 2018; Yuan et al., 2020; Hoe et al., 2021; Wang et al., 2023).Information-theoretic approaches such as the variational information bottleneck (Alemi et al., 2016) and mutual-information neural estimation (Belghazi et al., 2018) provide principled mechanisms for representation compression, and have begun to inform hashing methods—for example, via probabilistic modeling and hypothesis testing (Wang et al.). In addition, techniques such as knowledge distillation (Jang et al., 2022) have been utilized to obtain compact yet discriminative binary codes.Despite these advances, most existing hashing methods overlook fairness considerations. FATE (Zhang et al., 2024) makes the first attempt to incorporate fairness into deep hashing, but its interventions occur only in the final Hamming space. Due to the extreme compression and discreteness of binary codes, such post-hoc mitigation inherently limits the ability to disentangle and suppress sensitive factors without degrading semantic utility.

**Disentangled Representation Learning.** Disentangled representation learning(DRL) aims to encode underlying factors of variation into separate and interpretable dimensions of the latent space. Existing DRL methods can be broadly categorized into dimension-wise and vector-wise approaches based on their granularity of semantic alignment. Dimension-wise methods typically map one semantic factor to one latent dimension, e.g., $\beta$-VAE (Higgins et al., 2017), FactorVAE (Kim & Mnih, 2018), $\beta$-TCVAE (Chen et al., 2018), and GAN-based InfoGAN (Chen et al., 2016). By contrast, vector-wise methods represent a factor with a low-dimensional subspace, e.g., DR-GAN (Tran et al., 2017), DRNET (Denton et al., 2017), and MAP-IVR (Liu et al., 2021). Recent advancements integrate contrastive learning with disentanglement paradigms to enhance representation quality without relying heavily on labeled data (Li et al., 2021; Wang et al., 2024). In the context of fairness, disentanglement has been explored to isolate sensitive attributes from task-relevant features (Zhu et al., 2024; Zhang et al., 2025b), yielding improved fairness–utility trade-offs.

**Fairness in Machine Learning.** Fairness in machine learning has been extensively studied, particularly in classification and recommendation systems (Agarwal & Deshpande, 2022; Padh et al., 2021; Li et al., 2023). Common fairness notions include demographic parity, equalized odds and

individual fairness(Caton & Haas, 2024). Techniques to mitigate bias can be broadly categorized into pre-processing (e.g., data reweighting or transformation (Krasanakis et al., 2018; Gronowski et al., 2023)), in-processing (e.g., adversarial debiasing (Celis & Keswani, 2019; Xu et al., 2019) or fairness constraints (Donini et al., 2018)), and post-processing methods (e.g., calibration of outputs (Hébert-Johnson et al., 2018)). Despite this progress, fairness in retrieval systems—particularly hashing-based methods—remains under-explored. Recent work by Zhang et al. (2024) represents a notable step toward fair hashing by incorporating adversarial learning and contrastive objectives directly in the hash space. However, operating solely in the compressed and discrete hash space limits the flexibility and effectiveness of bias mitigation.

## 3 PROBLEM DEFINITION

Let $\mathcal{D} = \{I_i, y_i, s_i\}_{i=1}^N$ denote a dataset, where $I_i$ represents an input image, $y_i \in \mathcal{Y}$ is the corresponding target attribute label, and $s_i \in \mathcal{S}$ denotes the associated sensitive attribute label. The objective is to learn a deep hashing function $\mathcal{H} : I \mapsto \boldsymbol{b} \in \{-1, 1\}^L$, which maps images to compact binary hash codes of length $L$. This function should satisfy two properties: **(1) High Retrieval Accuracy**: relevant images (i.e., those sharing label $y_i$) should be ranked above those irrelevant ones; **(2) Group Fairness**: retrieval outcomes should be equitable across sensitive groups $\mathcal{S}$.

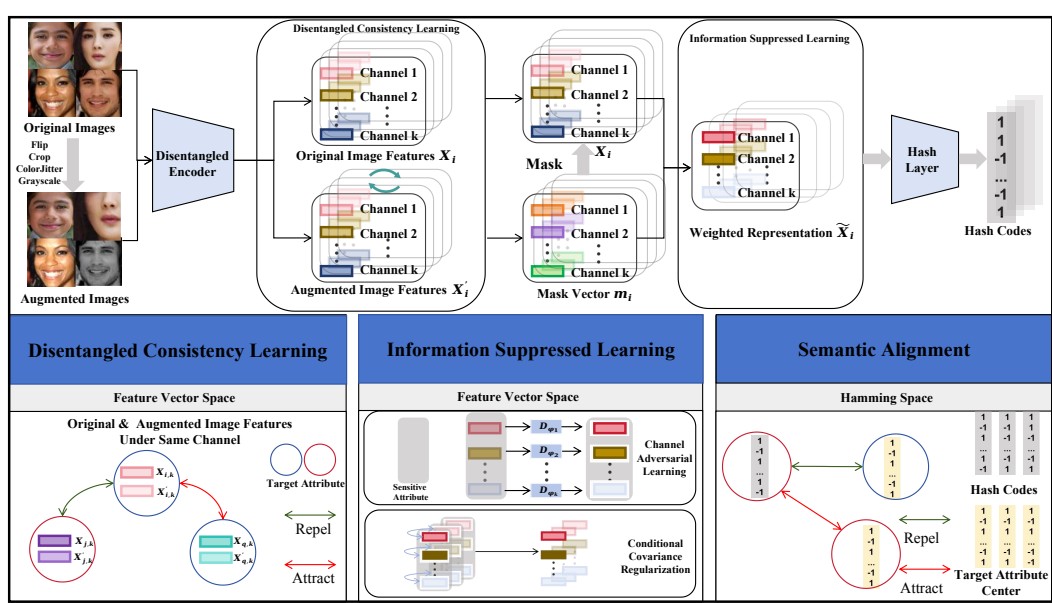

Figure 1: Overview of DISH. Here, $\mathbf{x}_{i,k}$ and $\mathbf{x}'_{i,k}$ are the $k$-th factor features for $I_i$ and $I'_i$; $\mathbf{m}_i = [p_\theta(k \mid \mathbf{x}_i)]_{k=1}^K$ is the assignment mask and $\tilde{\mathbf{x}}_{i,k} = m_{i,k}\mathbf{x}_{i,k}$ the weighted feature; $D_{\psi_k}$ (with GRL) is the channel discriminator; DISH learns fair and discriminative hash codes through disentangled consistency learning, information suppressed learning, and semantic alignment.

## 4 THE PROPOSED FRAMEWORK

This section presents the overall architecture of the proposed DISH framework, which comprises four primary components: (1) *Disentangled Encoder*, (2) *Disentangled Consistency Learning*, (3) *Information-Suppressed Learning*, and (4) *Semantic Alignment*. The first three components operate in the continuous representation space, whereas the fourth operates in the discrete Hamming space. A schematic of the full architecture is shown in Figure 1.

### 4.1 DISENTANGLED ENCODER

The Disentangled Encoder is designed to decompose each input image into $K$ factor-specific representations, enabling the inference of underlying latent factors that contribute to the image content.

The encoder comprises a pre-trained feature extractor (e.g., ResNet-50 (He et al., 2016)) to obtain high-level semantic features, followed by $K$ parallel multilayer perceptron (MLP) branches. Each branch is responsible for modeling a distinct and approximately independent factor of variation. Given an input image $I_i$, the encoder produces a disentangled feature representation: $\mathbf{x}_i = [\mathbf{x}_{i,1}, \mathbf{x}_{i,2}, \ldots, \mathbf{x}_{i,K}] \in \mathbb{R}^d$, where $\mathbf{x}_{i,k} \in \mathbb{R}^{d/K}$ ($1 \leq k \leq K$) corresponds to the $k$-th factor-specific component, and $d$ denotes the total feature dimension. Additionally, we apply data augmentation to $I_i$ (e.g., random cropping, flipping, and color jittering) to obtain an augmented view $I_i'$. The encoder processes $I_i'$ through the same network, yielding: $\mathbf{x}_i' = [\mathbf{x}_{i,1}', \mathbf{x}_{i,2}', \ldots, \mathbf{x}_{i,K}'] \in \mathbb{R}^d$, which serves as a counterpart for disentangled consistency learning.

## 4.2 DISENTANGLED CONSISTENCY LEARNING

We define "consistency" at the factor level: the model's responses to the same image and its augmented view should remain stable; samples from the same class should cluster while different classes separate; and the data-driven factor assignment should align with label-based semantic evidence. Crucially, the "factors" discovered here are not pre-defined attributes but latent discriminative components learned in a purely data driven manner. To achieve this, we perform supervised contrastive learning within each factor channel, using labels to shape the similarity structure so that semantic information concentrates in a small number of interpretable factors rather than being mixed in a single representation space. The prototype-based assignments are then coupled with the contrastive signal, encouraging stable posterior preferences.

Firstly, given $\mathbf{x}_i$, we compute the factor assignment probability $p_\theta(k \mid \mathbf{x}_i)$ using a prototype-based method. Specifically, we introduce $K$ latent factor prototypes $\{\mathbf{c}_k\}_{k=1}^K$. The probability that the $k^{\text{th}}$ latent factor is reflected in representation $\mathbf{x}_i$ is parameterized as:

$$p_\theta(k \mid \mathbf{x}_i) = \frac{\exp\big(\phi(\mathbf{x}_{i,k}, \mathbf{c}_k)\big)}{\sum_{k'=1}^K \exp\big(\phi(\mathbf{x}_{i,k'}, \mathbf{c}_{k'})\big)}. \tag{1}$$

where $\phi$ is the temperature-scaled cosine similarity and $\tau > 0$ is the temperature:

$$\phi(\mathbf{a}, \mathbf{b}) = \frac{\mathbf{a}^\top \mathbf{b}}{\|\mathbf{a}\|_2 \|\mathbf{b}\|_2 \tau}. \tag{2}$$

Then, we define the supervised contrastive learning task under $k$-th latent factor. Given a minibatch $\mathcal{B}$, define $A(i) = \mathcal{B} \setminus \{i\}$ and $P(i) = \{p \in A(i) : y_p = y_i\}$. The contrastive softmax likelihood is:

$$p_\theta(y_i \mid \mathbf{x}_i, k) = \frac{1}{|P(i)|} \sum_{p \in P(i)} \frac{\exp\big(\phi(\mathbf{x}_{i,k}, \mathbf{x}_{p,k}')\big)}{\sum_{a \in A(i)} \exp\big(\phi(\mathbf{x}_{i,k}, \mathbf{x}_{a,k}')\big)}. \tag{3}$$

We model the label evidence over latent factors as:

$$p_\theta(y_i \mid \mathbf{x}_i) = \sum_{k=1}^K p_\theta(k \mid \mathbf{x}_i) \, p_\theta(y_i \mid \mathbf{x}_i, k). \tag{4}$$

However, direct optimization is intractable due to the latent factors. Therefore, we instead optimize the evidence lower bound (ELBO) of the log-likelihood. For any distribution $q(k \mid \mathbf{x}_i, y_i)$:

$$\log p_\theta(y_i \mid \mathbf{x}_i) \geq \mathbb{E}_{k \sim q}[\log p_\theta(y_i \mid \mathbf{x}_i, k)] - D_{\mathrm{KL}}(q(\cdot \mid \mathbf{x}_i, y_i) \,\|\, p_\theta(\cdot \mid \mathbf{x}_i)). \tag{5}$$

Equality is attained by the variational posterior:

$$q_\theta(k \mid \mathbf{x}_i, y_i) = \frac{p_\theta(k \mid \mathbf{x}_i) \, p_\theta(y_i \mid \mathbf{x}_i, k)}{\sum_{k'=1}^K p_\theta(k' \mid \mathbf{x}_i) \, p_\theta(y_i \mid \mathbf{x}_i, k')}. \tag{6}$$

Apply Jensen's inequality to $\log \mathbb{E}_q[\cdot]$ to obtain equation 5. Maximization w.r.t. $q$ yields equation 6.

**Jensen Bound For Contrastive Likelihood.** Let $w_{i,k}(a) = \exp\{\phi(\mathbf{x}_{i,k}, \mathbf{x}_{a,k}')\}$ and $u_p = \frac{w_{i,k}(p)}{\sum_{a \in A(i)} w_{i,k}(a)} \in (0, 1)$. Then, by concavity of log, $\log \frac{1}{|P(i)|} \sum_{p \in P(i)} u_p \geq \frac{1}{|P(i)|} \sum_{p \in P(i)} \log u_p$:

$$\log p_\theta(y_i \mid \mathbf{x}_i, k) \geq \frac{1}{|P(i)|} \sum_{p \in P(i)} \Big( \log w_{i,k}(p) - \log \sum_{a \in A(i)} w_{i,k}(a) \Big). \tag{7}$$

**Computable Minibatch Lower Bound.** Define the per-factor term with a partition function

$$Z_{i,k} := \sum_{a \in A(i)} \exp\big(\phi(\mathbf{x}_{i,k}, \mathbf{x}'_{a,k})\big), \ell_{i,k} := \frac{1}{|P(i)|} \sum_{p \in P(i)} \Big(\phi(\mathbf{x}_{i,k}, \mathbf{x}'_{p,k}) - \log Z_{i,k}\Big). \quad (8)$$

Combining equation 5 and equation 7, for any $q$, we get the following inequality:

$$\log p_\theta(y_i \mid \mathbf{x}_i) \geq \sum_{k=1}^{K} q(k \mid \mathbf{x}_i, y_i) \, \ell_{i,k} \ - \ D_{\mathrm{KL}}\big(q(\cdot \mid \mathbf{x}_i, y_i) \,\|\, p_\theta(\cdot \mid \mathbf{x}_i)\big). \quad (9)$$

Let $q_\theta$ be the variational posterior in equation 6. Define the batch lower bound (to maximize)

$$\mathcal{L}_{\mathrm{DCL}}^{\mathrm{lb}} = \sum_{i \in \mathcal{B}} \Big( \sum_{k=1}^{K} q_\theta(k \mid \mathbf{x}_i, y_i) \, \ell_{i,k} - D_{\mathrm{KL}}\big(q_\theta \,\|\, p_\theta\big) \Big). \quad (10)$$

In practice, we optimize the negative of the bound as the training loss:

$$\mathcal{L}_{\mathrm{DCL}} = - \mathcal{L}_{\mathrm{DCL}}^{\mathrm{lb}}. \quad (11)$$

### 4.3 INFORMATION-SUPPRESSED LEARNING

To promote invariance to the sensitive attribute without sacrificing semantic discriminability, we introduce an information-suppressed framework that combines probability-driven channel masking, channel-wise adversarial learning, and conditional covariance regularization to preserve task semantics while removing both per-channel and cross-channel sensitive cues.

**Channel Masking.** Building upon the factor assignment probabilities $p_\theta(k \mid \mathbf{x}_i)$ from equation 1, we design a probability-driven masking concentrates information into high-assignment factors and attenuates low-assignment ones, without requiring access to target labels or sensitive attributes. For each representation $\mathbf{x}_i$, we compute a channel-wise mask vector.

$$\mathbf{m}_i = \big[m_{i,1}, \ldots, m_{i,K}\big] = \big[p_\theta(1 \mid \mathbf{x}_i), \ldots, p_\theta(K \mid \mathbf{x}_i)\big]. \quad (12)$$

We then obtain an assignment weighted representation via channel-wise scaling:

$$\tilde{\mathbf{x}}_i = \mathbf{x}_i \odot_c \mathbf{m}_i = \big[\mathbf{x}_{i,1} \, m_{i,1}, \ldots, \mathbf{x}_{i,K} \, m_{i,K}\big]. \quad (13)$$

where $\odot_c$ denotes multiplication by channel. This multiplication attenuates channels with lower $p_\theta(k \mid \mathbf{x}_i)$, concentrating semantics into more informative factors.

**Channel Adversarial Learning.** Let $s_i \in \{1, \ldots, C_s\}$ denote the sensitive attribute of $I_i$. From equation 13, write $\tilde{\mathbf{x}}_i = \big[\tilde{\mathbf{x}}_{i,1}, \ldots, \tilde{\mathbf{x}}_{i,K}\big]$ with $\tilde{\mathbf{x}}_{i,k} = m_{i,k} \mathbf{x}_{i,k} \in \mathbb{R}^{d/K}$ the $k$-th channel feature. For each channel, we instantiate a discriminator $D_{\psi_k} : \mathbb{R}^{d/K} \to \Delta^{C_s - 1}$ that predicts sensitive label from $\tilde{\mathbf{x}}_{i,k}$. We define adversarial loss by averaging the per-channel cross-entropy over minibatch:

$$\mathcal{L}_{\mathrm{CAL}}(\theta, \{\psi_k\}_{k=1}^K) = \frac{1}{|\mathcal{B}| \, K} \sum_{i \in \mathcal{B}} \sum_{k=1}^{K} \mathrm{CE}\big(s_i, \, D_{\psi_k}(\tilde{\mathbf{x}}_{i,k}(\theta))\big). \quad (14)$$

where CE denotes cross-entropy. Fairness is enforced via the following saddle-point objective:

$$\min_{\{\psi_k\}_{k=1}^K} \max_\theta \ \mathcal{L}_{\mathrm{CAL}}(\theta, \{\psi_k\}_{k=1}^K). \quad (15)$$

In practice, we implement the saddle-point objective in equation 15 via a gradient-reversal layer (GRL), which preserves the theoretical min–max view while yielding a simple training procedure. The GRL encourages each channel feature $\tilde{\mathbf{x}}_{i,k}$ to be uninformative about $s_i$. This realization is theoretically consistent with the per-channel min–max scheme and empirically stable in training.

**Conditional Covariance Regularization.** While the adversarial objective suppresses direct leakage of $S$ from each channel, it does not prevent channels from jointly encoding complementary sensitive information. To further mitigate this effect, we introduce a conditional covariance penalty that encourages approximate conditional independence between channels given $S$. In practice, for a minibatch $\mathcal{B}$ with representations $\{\tilde{\mathbf{x}}_i\}_{i \in \mathcal{B}}$ and corresponding sensitive labels $\{s_i\}_{i \in \mathcal{B}}$, we first group

samples by their sensitive class. For each class $s$, let $\mathcal{B}_s = \{i \in \mathcal{B} : s_i = s\}$ denote the subset of samples with label $s$. For each pair of channels $(k, \ell)$, we compute the empirical covariance:

$$\mathrm{Cov}\big(\tilde{\mathbf{X}}_k, \tilde{\mathbf{X}}_\ell \mid S = s\big) = \frac{1}{|\mathcal{B}_s| - 1} \sum_{i \in \mathcal{B}_s} \big(\tilde{\mathbf{x}}_{i,k} - \bar{\mathbf{x}}_{k,s}\big)\big(\tilde{\mathbf{x}}_{i,\ell} - \bar{\mathbf{x}}_{\ell,s}\big)^\top. \tag{16}$$

where $\bar{\mathbf{x}}_{k,s} = \frac{1}{|\mathcal{B}_s|} \sum_{i \in \mathcal{B}_s} \tilde{\mathbf{x}}_{i,k}$. The conditional covariance regularization loss is then defined as

$$\mathcal{L}_{\mathrm{CCR}}(\theta) = \sum_{s=1}^{C_s} \sum_{k \neq \ell} \big\| \mathrm{Cov}\big(\tilde{\mathbf{X}}_k, \tilde{\mathbf{X}}_\ell \mid S = s\big) \big\|_F^2. \tag{17}$$

This conditional covariance regularizer promotes conditional decorrelation between channels, reducing the possibility that sensitive information is recoverable through higher-order interactions.

**Theoretical Properties.** We provide an information-theoretic characterization of the channel-wise adversarial objective in equation 14. Let $S$ be the sensitive attribute with entropy $H(S)$, and let $\tilde{\mathbf{X}}_k$ denote the random variable corresponding to the $k$-th masked channel feature $\tilde{\mathbf{x}}_{i,k}$. For fixed encoder parameters $\theta$, minimizing the per-channel cross-entropy yields the Bayes discriminator $D^*_{\psi_k}(\tilde{\mathbf{x}}_{i,k}) = p_\theta(s_i \mid \tilde{\mathbf{x}}_{i,k})$, and the inner optimum equals the average conditional entropy:

$$\min_{\{\psi_k\}} \mathcal{L}_{\mathrm{CAL}}(\theta, \{\psi_k\}) = \frac{1}{K} \sum_{k=1}^K H\big(S \big| \tilde{\mathbf{X}}_k\big). \tag{18}$$

The outer maximization over $\theta$ is equivalent to minimizing the average mutual information.

$$\max_\theta \frac{1}{K} \sum_k H(S \mid \tilde{\mathbf{X}}_k) \iff \min_\theta \frac{1}{K} \sum_k I_\theta\big(S; \tilde{\mathbf{X}}_k\big). \tag{19}$$

since $I_\theta(S; \tilde{\mathbf{X}}_k) = H(S) - H(S \mid \tilde{\mathbf{X}}_k)$. Let $\tilde{\mathbf{X}} = [\tilde{\mathbf{X}}_1, \ldots, \tilde{\mathbf{X}}_K]$ be the concatenated masked representation; by the chain rule of mutual information, $I_\theta(S; \tilde{\mathbf{X}}) = \sum_k I_\theta(S; \tilde{\mathbf{X}}_k \mid \tilde{\mathbf{X}}_{1:(k-1)}) \leq \sum_k I_\theta(S; \tilde{\mathbf{X}}_k)$. Under the mild condition that $\{\tilde{\mathbf{X}}_k\}$ are approximately conditionally uncorrelated given $S$ (enforced in practice by minimizing $\sum_{k \neq \ell} \|\mathrm{Cov}(\tilde{\mathbf{X}}_k, \tilde{\mathbf{X}}_\ell \mid S)\|_F^2$), the upper bound becomes tight, so minimizing $\frac{1}{K} \sum_k I_\theta(S; \tilde{\mathbf{X}}_k)$ effectively reduces global leakage $I_\theta(S; \tilde{\mathbf{X}})$. For any downstream hash mapping $\mathbf{B} = h(\tilde{\mathbf{X}}) \in \{-1, +1\}^L$, data processing further gives $I_\theta(S; \mathbf{B}) \leq I_\theta(S; \tilde{\mathbf{X}}) \leq \sum_k I_\theta(S; \tilde{\mathbf{X}}_k)$; moreover, for any bounded retrieval score $g : \{-1, +1\}^L \to [0, 1]$, Pinsker-type inequalities imply $\big|\mathbb{E}[g(\mathbf{B}) \mid S{=}a] - \mathbb{E}[g(\mathbf{B}) \mid S{=}b]\big| \leq 2\,\mathrm{TV}(P_{\mathbf{B}|S=a}, P_{\mathbf{B}|S=b}) \leq C\sqrt{I_\theta(S; \mathbf{B})} \leq C\sqrt{\sum_k I_\theta(S; \tilde{\mathbf{X}}_k)}$ for a universal constant $C > 0$. Thus, reducing the channel-averaged leakage tightens an explicit, information-theoretic bound on downstream retrieval.

### 4.4 SEMANTIC ALIGNMENT

We generate discrete hash codes in the Hamming space from masked representations. Let $f_\eta : \mathbb{R}^d \to \mathbb{R}^L$ be the hash head with parameters $\eta$. Given $\tilde{\mathbf{x}}_i$ from equation 13, we compute

$$\mathbf{u}_i = f_\eta(\tilde{\mathbf{x}}_i), \qquad \mathbf{b}_i = \mathrm{sign}(\mathbf{u}_i) \in \{-1, +1\}^L. \tag{20}$$

and use a differentiable relaxation $\mathbf{r}_i = \tanh(\mathbf{u}_i)$ for backpropagation during training. To inject class semantics, we maintain $C = |\mathcal{Y}|$ *semantic anchors* $\{\mathbf{z}_c\}_{c=1}^C$ in the Hamming space, each $\mathbf{z}_c \in \{-1, +1\}^L$. Anchors are initialized by i.i.d. Rademacher sampling (each bit $\pm 1$ with probability $1/2$), which yields an expected inter-class Hamming distance of $L/2$ and thus large initial separation. The semantic alignment loss is defined as follows, where $\tau_s > 0$ is a temperature.

$$\mathcal{L}_{\mathrm{SA}}(\theta, \eta, \{\mathbf{z}_c\}) = -\sum_{i \in \mathcal{B}} \log \frac{\exp\big(\mathbf{z}_{y_i}^\top \mathbf{r}_i / \tau_s\big)}{\sum_{c=1}^C \exp\big(\mathbf{z}_c^\top \mathbf{r}_i / \tau_s\big)}. \tag{21}$$

### 4.5 OVERALL OBJECTIVE

Let $\theta$ denote the encoder and prototype parameters, $\Psi = \{\psi_k\}_{k=1}^K$ the per-channel discriminators, and $\eta$ the hash head. We adopt a single saddle-point formulation that unifies disentangled consistency, channel adversarial learning, conditional covariance regularization, and semantic alignment:

$$\min_{\theta,\eta,\{\mathbf{z}_c\}} \max_{\{\psi_k\}} \left[ \underbrace{\mathcal{L}_{\text{DCL}}(\theta)}_{\substack{\text{disentangled consistency} \\ \text{(min)}}} - \lambda_1 \underbrace{\mathcal{L}_{\text{CAL}}(\theta, \{\psi_k\})}_{\substack{\text{channel adversary} \\ \text{(max by } \theta\text{, min by } \psi_k\text{)}}} + \lambda_2 \underbrace{\mathcal{L}_{\text{CCR}}(\theta)}_{\substack{\text{conditional covariance} \\ \text{(min)}}} + \underbrace{\mathcal{L}_{\text{SA}}(\theta, \eta, \{\mathbf{z}_c\})}_{\substack{\text{semantic alignment} \\ \text{(min)}}} \right]. \quad (22)$$

Here, $\lambda_1 > 0$ controls the strength of channel adversarial learning and $\lambda_2 > 0$ controls the strength of conditional covariance regularization. $\mathcal{L}_{\text{DCL}}(\theta)$ is the *disentangled consistency* loss (Eq. 11), encouraging factor assignments. $\mathcal{L}_{\text{CAL}}(\theta, \{\psi_k\})$ is the *channel adversarial* loss (Eq. 14), where each discriminator $\psi_k$ is trained to minimize sensitive prediction error while the encoder maximizes it, thus suppressing recoverable sensitive information in masked factors. $\mathcal{L}_{\text{CCR}}(\theta)$ is the *conditional covariance regularization* loss (Eq. 17), penalizing cross-channel covariance given sensitive labels to reduce joint sensitive leakage. Finally, $\mathcal{L}_{\text{SA}}(\theta, \eta, \{\mathbf{z}_c\})$ is the *semantic alignment* loss (Eq. 21), pulling hash logits in Hamming space. We provide an algorithm procedure in the appendix B.

## 5 EXPERIMENT

### 5.1 DATASET

We evaluated on two facial attribute datasets, UTKFace (Zhang et al., 2017) and CelebA (Liu et al., 2015). UTKFace ($\approx$ 20K images with age/gender/ethnicity annotations) is used in two configurations. In the first, *Ethnicity* is the target label (five categories) and *Age* serves as the sensitive attribute, binarized as $<35$ vs. $\geq35$. In the second, *Age* becomes the target (five bins: 0–20, 20–40, 40–60, 60–80, 80+) and *Ethnicity* is the sensitive attribute, binarized as European American vs. non–European American. CelebA ($\approx$ 200K images with 40 binary attributes) is used with *Attractive* as target and *Male* as the sensitive attribute. For retrieval evaluation, we randomly draw 100 query images on UTKFace and 500 on CelebA; all remaining images are used for training and retrieval.

### 5.2 EVALUATION METRICS

We evaluate two aspects: retrieval accuracy and group fairness.
**Retrieval accuracy.** *Mean Average Precision*: $\text{MAP} = \frac{1}{|Q|} \sum_{q \in Q} \frac{1}{m_q} \sum_{k=1}^{n_q} \text{Precision@}k(q) \cdot \text{rel}_q(k)$, where $Q$ is the query set, $m_q$ is the number of relevant items for query $q$, $n_q$ is the list length, and $\text{rel}_q(k) \in \{0, 1\}$ indicates relevance at rank $k$.
**Fairness Metrics.** We use DP, EOP, and EOD as our fairness measures.
*Demographic Parity (DP)*: $\text{DP} = \left| P(\hat{Q}_i \mid S_i{=}1) - P(\hat{Q} \mid S_i{=}0) \right|$

*Equal Opportunity (EOP)*: $\text{EOP} = \left| P(\hat{Q}_i \mid Y_i{=}1, S_i{=}1) - P(\hat{Q}_i \mid Y_i{=}1, S_i{=}0) \right|$

*Equalized Odds (EOD)*: $\text{EOD} = \left| P(\hat{Q}_i \mid Y_i{=}y, S_i{=}1) - P(\hat{Q}_i \mid Y_i{=}y, S_i{=}0) \right|_{y \in \{0,1\}}$

### 5.3 PERFORMANCE COMPARISON

We conduct extensive experiments on three benchmark datasets (UTKFace with two target–sensitive configurations and CelebA) under hash code lengths ranging from 16 to 128 bits, and compare against a diverse set of competitive baselines; the full list is provided in appendix C. Each experiment is repeated five times with different random seeds, and we report the mean and standard deviation of all metrics to ensure statistical robustness. The results consistently show that DISH achieves the best performance in terms of both retrieval accuracy (MAP) and fairness measures (DP, EOP, and EOD), thereby representing a clear Pareto improvement over existing baselines. For example, on UTKFace with ethnicity as the target attribute, DISH reaches a MAP of 72.99 at 16 bits while simultaneously reducing EOP and EOD to 2.08 and 4.22, respectively. Similar trends hold across longer code lengths, the alternative UTKFace setting with age as target, and the CelebA dataset. These consistent gains can be attributed to the design of our framework: disentangled consistency learning ensures

that semantic information is stably concentrated within factor-specific channels; probability-driven channel masking together with channel-wise adversarial learning effectively suppresses sensitive leakage at the per-channel level; and conditional covariance regularization further mitigates cross-channel correlations that could reintroduce bias. Finally, semantic alignment in the Hamming space preserves inter-class discriminability after binarization.

Table 1: Performance comparison (%) with the state-of-the-art methods on UTKFace with code lengths varying from 16 to 128. Target Attribute: ethnicity, Sensitive Attribute: age.

| Method | 16 bits | | | | 32 bits | | | | 64 bits | | | | 128 bits | | | |
|---|---|---|---|---|---|---|---|---|---|---|---|---|---|---|---|---|
| | MAP↑ | EOD↓ | EOP↓ | DP↓ | MAP↑ | EOD↓ | EOP↓ | DP↓ | MAP↑ | EOD↓ | EOP↓ | DP↓ | MAP↑ | EOD↓ | EOP↓ | DP↓ |
| OrthoHash | 57.92±1.04 | 11.03±0.60 | 6.83±0.22 | 7.43±0.95 | 59.34±1.30 | 11.30±1.45 | 7.08±0.31 | 7.81±1.28 | 61.90±2.45 | 10.95±1.57 | 6.92±0.47 | 7.89±1.29 | 62.79±3.10 | 10.57±1.01 | 6.50±0.46 | 8.18±0.41 |
| Bihalf | 52.35±1.63 | 13.88±2.27 | 9.20±0.68 | 7.24±1.19 | 55.26±1.10 | 11.33±1.76 | 7.36±1.04 | 7.28±1.33 | 56.93±1.49 | 11.00±1.76 | 6.87±0.71 | 7.50±1.93 | 55.94±1.10 | 11.28±1.82 | 7.17±0.86 | 7.44±1.48 |
| CE | 55.62±2.37 | 10.48±0.80 | 7.03±0.42 | 6.01±0.81 | 58.58±1.38 | 10.02±1.32 | 6.76±0.58 | 6.76±1.00 | 59.35±1.33 | 11.19±0.68 | 7.55±0.34 | 7.37±0.68 | 60.05±0.87 | 10.57±1.06 | 7.02±0.55 | 7.12±0.42 |
| CSQ | 63.14±1.25 | 9.41±1.50 | 5.35±0.37 | 7.92±0.44 | 64.23±1.14 | 9.01±0.42 | 5.19±0.08 | 7.91±0.44 | 65.27±1.02 | 8.68±0.82 | 5.16±0.29 | 7.75±0.21 | 62.14±3.89 | 9.15±1.38 | 5.51±0.38 | 7.50±1.03 |
| DFH | 46.74±2.75 | 14.99±1.79 | 8.23±0.77 | 9.27±1.44 | 55.07±2.59 | 13.75±3.01 | 7.97±1.23 | 8.79±1.95 | 59.20±3.07 | 14.12±1.23 | 7.92±0.39 | 9.12±1.17 | 60.64±2.87 | 16.40±4.21 | 10.32±1.66 | 9.60±0.85 |
| DPSH | 62.30±1.13 | 9.47±0.87 | 5.41±0.17 | 8.11±0.56 | 62.99±0.59 | 10.27±0.66 | 6.03±0.25 | 8.48±0.25 | 64.12±0.46 | 10.50±0.69 | 6.33±0.11 | 8.59±0.42 | 63.71±1.24 | 10.19±1.00 | 6.09±0.23 | 8.43±0.61 |
| DTSH | 60.83±1.63 | 9.09±0.40 | 5.37±0.23 | 7.74±0.28 | 61.53±1.97 | 11.91±1.33 | 7.30±0.28 | 8.89±1.92 | 60.83±0.38 | 11.93±1.45 | 7.63±0.19 | 8.69±0.98 | 60.61±1.74 | 10.94±1.12 | 7.06±0.27 | 8.01±0.40 |
| GreedyHash | 64.62±1.23 | 10.08±1.39 | 6.51±0.50 | 7.87±0.62 | 64.63±1.41 | 10.42±0.60 | 7.22±0.40 | 7.70±0.41 | 63.78±0.78 | 10.63±0.66 | 7.30±0.40 | 7.74±0.46 | 55.92±9.19 | 10.60±1.22 | 6.08±0.87 | 8.26±0.94 |
| SDH-C | 57.48±1.51 | 10.15±2.17 | 6.13±0.13 | 6.73±1.72 | 62.77±0.52 | 11.78±1.69 | 7.81±0.28 | 8.29±0.78 | 63.48±0.32 | 11.82±0.38 | 8.18±0.23 | 7.75±0.35 | 63.98±1.31 | 12.25±0.83 | 8.65±0.22 | 7.57±0.12 |
| DLBD | 29.17±1.26 | 17.39±1.29 | 10.16±0.33 | 7.79±0.51 | 29.22±1.26 | 17.47±1.29 | 10.35±0.33 | 7.71±0.41 | | 16.51±1.17 | 9.79±1.19 | 7.31±0.44 | 31.44±1.13 | 17.50±1.94 | 10.31±0.93 | 7.71±1.00 |
| MDSHC | 62.88±1.66 | 8.64±1.13 | 5.96±0.67 | 7.04±0.32 | 65.75±1.65 | 7.77±0.53 | 5.16±0.28 | 6.69±0.31 | 64.11±1.33 | 8.84±0.33 | 6.69±0.89 | 6.90±0.15 | 62.78±0.68 | 8.62±1.14 | 5.39±0.25 | 7.76±0.41 |
| FATE | 70.01±1.27 | 4.47±1.26 | 2.57±0.52 | 5.09±0.32 | 69.98±1.18 | 5.37±0.64 | 3.96±0.28 | 6.60±0.33 | 72.22±1.46 | 6.12±0.44 | 3.39±0.42 | 6.03±0.75 | 72.62±1.18 | 6.51±0.54 | 3.68±0.35 | 6.42±0.31 |
| **DISH** | **72.99±1.32** | **4.22±0.75** | **2.08±0.75** | **4.79±0.15** | **73.20±1.64** | **4.12±0.57** | **2.55±0.49** | **6.54±0.28** | **73.67±1.83** | **4.28±0.98** | **2.96±0.84** | **5.63±0.37** | **73.33±0.91** | **3.66±0.64** | **2.37±0.45** | **6.10±0.26** |

Table 2: Performance comparison (%) with the state-of-the-art methods on UTKFace with code lengths varying from 16 to 128. Target Attribute: age, Sensitive Attribute: ethnicity.

| Method | 16 bits | | | | 32 bits | | | | 64 bits | | | | 128 bits | | | |
|---|---|---|---|---|---|---|---|---|---|---|---|---|---|---|---|---|
| | MAP↑ | EOD↓ | EOP↓ | DP↓ | MAP↑ | EOD↓ | EOP↓ | DP↓ | MAP↑ | EOD↓ | EOP↓ | DP↓ | MAP↑ | EOD↓ | EOP↓ | DP↓ |
| OrthoHash | 56.41±1.48 | 12.97±1.48 | 4.87±0.27 | 11.08±0.65 | 58.48±0.89 | 14.91±2.19 | 5.76±0.14 | 12.45±1.55 | 59.35±0.19 | 15.20±1.43 | 5.93±0.09 | 12.98±0.70 | 60.64±0.86 | 14.80±0.84 | 5.60±0.3 | 11.60±0.41 |
| Bihalf | 52.55±2.2 | 19.97±4.56 | 7.56±0.37 | 15.07±2.86 | 53.30±2.33 | 18.88±3.14 | 6.78±1.45 | 14.42±1.31 | 54.46±1.2 | 18.84±0.7 | 6.59±0.35 | 14.46±0.55 | 53.58±1.15 | 17.53±0.92 | 6.60±0.38 | 12.57±0.82 |
| CE | 57.87±1.5 | 15.66±0.58 | 5.62±0.67 | 12.90±0.79 | 57.08±0.86 | 15.15±0.58 | 5.28±0.22 | 12.29±0.39 | 57.93±1.6 | 16.02±1.15 | 5.79±0.56 | 12.74±0.55 | 57.67±0.61 | 15.03±0.88 | 5.23±0.28 | 12.37±0.51 |
| CSQ | 62.11±1.37 | 12.44±1.35 | 4.18±0.79 | 11.75±1.54 | 64.16±1.06 | 12.87±0.78 | 4.82±0.48 | 11.89±0.53 | 64.61±1.56 | 11.82±0.97 | 4.48±0.23 | 10.93±0.53 | 64.73±1.22 | 12.18±0.5 | 4.28±0.13 | 13.46±0.56 |
| DFH | 56.62±1.24 | 13.72±0.94 | 4.35±0.61 | 12.43±0.99 | 55.91±1.66 | 14.98±0.55 | 5.60±0.36 | 12.38±0.47 | 57.72±1.17 | 15.81±2.13 | 6.12±0.54 | 12.81±1.15 | 57.60±1.39 | 16.75±1.04 | 6.43±0.62 | 15.29±1.00 |
| DPSH | 57.64±1.51 | 16.37±0.48 | 5.93±0.55 | 14.18±0.43 | 56.84±1.11 | 17.27±1.4 | 6.32±0.36 | 14.78±0.92 | 57.46±1.01 | 17.15±0.44 | 6.35±0.42 | 14.40±0.28 | 58.15±0.41 | 17.01±0.55 | 6.11±0.41 | 15.17±0.64 |
| DTSH | 58.04±0.87 | 15.96±1.44 | 5.75±0.27 | 13.83±0.74 | 58.08±1.15 | 16.33±0.73 | 5.79±0.26 | 14.20±0.52 | 56.73±1.17 | 16.37±0.52 | 5.81±0.17 | 14.14±0.33 | 56.76±1.27 | 16.80±1.55 | 5.84±0.29 | 14.22±0.72 |
| GreedyHash | 61.35±1.89 | 14.99±0.7 | 5.32±0.12 | 12.40±0.2 | 61.59±1.05 | 15.64±1.04 | 5.91±0.4 | 12.71±0.69 | 60.68±0.56 | 14.90±0.43 | 5.51±0.36 | 12.15±0.28 | 58.96±0.64 | 14.95±0.68 | 5.77±0.15 | 11.74±0.63 |
| SDH-C | 50.36±1.13 | 13.77±2.93 | 4.74±0.92 | 11.28±1.35 | 59.75±1.61 | 16.17±0.53 | 6.04±0.12 | 12.58±0.47 | 60.21±1.22 | 16.22±0.05 | 6.10±0.21 | 12.22±0.25 | 60.69±0.75 | 15.84±0.43 | 2.94±0.28 | 11.87±0.41 |
| DLBD | 35.62±1.92 | 20.99±1.20 | 11.12±1.96 | 10.70±0.51 | 39.05±1.52 | 20.83±2.12 | 11.19±0.05 | 10.98±2.25 | 39.00±1.21 | 21.90±1.61 | 13.15±0.22 | 9.93±1.53 | 40.24±0.92 | 22.69±1.44 | 12.87±0.43 | 11.24±2.19 |
| MDSHC | 64.59±1.28 | 14.45±0.63 | 5.48±0.57 | 9.18±0.35 | 60.47±1.19 | 13.10±0.19 | 5.92±0.40 | 9.34±0.29 | 61.08±0.41 | 11.64±0.51 | 5.35±0.42 | 11.55±0.13 | 61.73±0.46 | 11.85±0.45 | 5.11±0.37 | 11.26±0.59 |
| FATE | 69.78±1.76 | 7.74±0.73 | 2.06±0.76 | 8.18±0.18 | 69.52±1.36 | 7.26±0.33 | 2.54±0.31 | 8.43±0.25 | 68.61±1.74 | 9.11±0.56 | 2.86±0.14 | 10.31±0.35 | 70.17±1.13 | 7.92±0.32 | 2.32±0.25 | 9.35±0.41 |
| **DISH** | **72.08±0.71** | **6.46±0.49** | **1.41±0.38** | **7.08±0.39** | **71.85±1.82** | **6.76±0.24** | **1.59±0.22** | **8.14±0.26** | **71.59±0.51** | **6.03±0.35** | **1.29±0.20** | **8.83±0.38** | **72.40±1.35** | **6.34±0.68** | **2.09±0.32** | **8.51±0.52** |

Table 3: Performance comparison (%) with the state-of-the-art methods on CelebA with code lengths varying from 16 to 128. Target Attribute: attractiveness, Sensitive Attribute: male.

| Method | 16 bits | | | | 32 bits | | | | 64 bits | | | | 128 bits | | | |
|---|---|---|---|---|---|---|---|---|---|---|---|---|---|---|---|---|
| | MAP↑ | EOD↓ | EOP↓ | DP↓ | MAP↑ | EOD↓ | EOP↓ | DP↓ | MAP↑ | EOD↓ | EOP↓ | DP↓ | MAP↑ | EOD↓ | EOP↓ | DP↓ |
| OrthoHash | 76.82±0.52 | 4.09±0.05 | 2.87±0.23 | 2.99±0.22 | 77.91±0.06 | 4.97±0.40 | 3.59±0.38 | 3.35±0.15 | 78.25±0.90 | 4.69±0.33 | 3.45±0.28 | 3.38±0.14 | 78.90±0.66 | 3.52±1.38 | 2.69±0.96 | 3.03±0.42 |
| Bihalf | 77.87±0.49 | 3.19±1.09 | 2.35±0.78 | 2.66±0.11 | 78.69±0.23 | 2.91±0.63 | 2.25±0.54 | 2.71±0.21 | 78.09±0.37 | 3.51±0.22 | 3.40±0.15 | 3.66±0.15 | 78.34±0.76 | 3.19±0.28 | 2.38±0.18 | 2.54±0.15 |
| CE | 77.33±0.42 | 2.98±0.68 | 2.70±0.56 | 2.40±0.13 | 78.10±0.93 | 3.88±0.74 | 2.88±0.56 | 2.90±0.33 | 77.27±0.13 | 4.56±0.35 | 3.35±0.28 | 3.13±0.11 | 77.97±0.58 | 4.76±0.14 | 3.54±0.18 | 3.18±0.13 |
| CSQ | 78.66±1.02 | 2.93±0.49 | 2.45±0.34 | 2.70±0.18 | 78.27±1.59 | 2.98±0.55 | 2.48±0.35 | 2.71±0.23 | 77.92±0.18 | 3.76±0.45 | 3.37±0.35 | 2.62±0.16 | 77.92±2.31 | 2.91±0.39 | 2.46±0.22 | 2.70±0.16 |
| DFH | 78.71±0.88 | 2.88±0.11 | 2.42±0.11 | 2.66±0.05 | 78.89±0.73 | 2.94±0.11 | 2.56±0.21 | 2.74±0.11 | 77.29±1.04 | 3.83±0.24 | 2.92±0.32 | 2.63±0.22 | 77.02±0.18 | 3.47±0.19 | 2.46±0.19 | 2.31±0.12 |
| DPSH | 78.37±0.96 | 4.33±1.25 | 3.22±0.79 | 2.85±0.67 | 79.01±0.86 | 2.92±0.98 | 2.21±0.74 | 2.46±0.16 | 77.47±0.04 | 5.09±1.94 | 3.80±1.23 | 3.15±0.72 | 78.37±2.95 | 4.84±1.57 | 3.59±1.42 | 2.94±0.74 |
| DTSH | 78.69±0.51 | 3.35±0.12 | 2.52±0.21 | 2.35±0.04 | 78.28±0.88 | 3.25±0.50 | 2.46±0.39 | 2.48±0.15 | 77.23±1.05 | 3.31±0.59 | 2.52±0.58 | 2.42±0.31 | 77.70±2.01 | 3.58±0.22 | 2.58±0.01 | 2.46±0.06 |
| GreedyHash | 78.62±0.89 | 3.18±0.31 | 2.83±0.11 | 2.53±0.14 | 77.74±1.24 | 3.52±0.26 | 2.57±0.11 | 2.37±0.12 | 77.57±1.19 | 3.73±0.65 | 2.70±0.37 | 2.54±0.16 | 77.35±0.78 | 4.06±0.43 | 2.94±0.28 | 2.71±0.06 |
| SDH-C | 78.79±0.79 | 3.09±0.19 | 2.33±0.17 | 2.54±0.17 | 78.37±0.15 | 3.58±0.24 | 2.55±0.22 | 2.54±0.32 | 78.05±0.26 | 3.34±0.11 | 2.54±0.12 | 2.41±0.14 | 77.23±0.81 | 3.37±0.39 | 2.43±0.23 | 2.40±0.18 |
| DLBD | 66.91±0.43 | 5.98±0.16 | 3.61±0.10 | 3.16±0.07 | 68.10±0.43 | 6.68±0.29 | 4.04±0.19 | 3.54±0.13 | 68.76±0.46 | 6.84±0.33 | 4.16±0.20 | 3.64±0.15 | 68.53±0.30 | 7.06±0.36 | 4.28±0.21 | 3.74±0.16 |
| MDSHC | 77.14±2.65 | 3.40±0.52 | 2.42±0.30 | 2.38±0.21 | 76.30±2.21 | 3.78±0.76 | 2.63±0.46 | 2.52±0.28 | 75.39±1.70 | 4.35±0.86 | 3.04±0.56 | 2.60±0.39 | 75.61±1.16 | 5.37±0.48 | 3.73±0.30 | 3.12±0.21 |
| FATE | 76.99±0.76 | 2.40±0.25 | 2.26±0.20 | 2.07±0.07 | 76.07±2.43 | 2.52±0.51 | 1.75±0.35 | 2.01±0.14 | 74.61±1.74 | 3.71±0.56 | 2.86±0.14 | 2.31±0.35 | 75.95±0.83 | 5.31±0.27 | 3.93±0.23 | 3.42±0.12 |
| **DISH** | **78.92±0.48** | **2.08±0.22** | **2.06±0.17** | **1.91±0.06** | **79.26±1.30** | **2.08±0.21** | **1.68±0.12** | **1.82±0.11** | **78.29±0.23** | **3.23±0.18** | **2.48±0.16** | **1.91±0.04** | **79.10±0.29** | **2.82±0.25** | **2.23±0.18** | **2.16±0.13** |

## 5.4 ABLATION STUDY

To further validate the effectiveness of different components in the proposed DISH framework, we conduct a series of ablation studies on the UTKFace dataset with 128-bit hash codes. Each experiment is repeated five times under different random seeds, and we report both mean and standard deviation to ensure statistical reliability. Specifically, we examine the contribution of each loss function, compare different masking strategies, and evaluate several model variants to better understand how the architectural choices affect retrieval accuracy and fairness.

Table 4: Performance comparison (%) with different masking methods with 128bits on UTKFace

| $\mathcal{L}_{DCL}$ | $\mathcal{L}_{CAL}$ | $\mathcal{L}_{CCR}$ | TA: ethnicity, SA: age | | | | TA: age, SA: ethnicity | | | |
|---|---|---|---|---|---|---|---|---|---|---|
| | | | MAP ↑ | EOD ↓ | EOP ↓ | DP ↓ | MAP ↑ | EOD ↓ | EOP ↓ | DP ↓ |
| - | - | - | 61.57 ± 1.25 | 10.73 ± 1.43 | 7.71 ± 0.64 | 7.84 ± 0.83 | 59.10 ± 1.27 | 14.01 ± 0.67 | 5.89 ± 0.44 | 11.06 ± 0.97 |
| ✓ | | | 62.14 ± 1.21 | 9.93 ± 0.87 | 7.70 ± 0.61 | 7.06 ± 0.54 | 61.39 ± 0.73 | 12.82 ± 1.13 | 5.36 ± 0.63 | 10.64 ± 0.72 |
| | ✓ | | 64.37 ± 1.78 | 8.95 ± 0.61 | 6.62 ± 0.54 | 6.77 ± 0.44 | 64.63 ± 1.00 | 10.77 ± 0.93 | 4.70 ± 0.68 | 9.41 ± 0.50 |
| | | ✓ | 64.58 ± 2.38 | 9.24 ± 1.84 | 5.74 ± 1.25 | 8.53 ± 0.90 | 67.82 ± 2.37 | 11.78 ± 1.54 | 3.76 ± 0.90 | 11.59 ± 0.67 |
| ✓ | ✓ | | 68.07 ± 0.66 | 7.23 ± 0.18 | 3.10 ± 0.15 | 8.43 ± 0.13 | 70.52 ± 1.03 | 6.84 ± 0.06 | 2.36 ± 0.11 | 8.92 ± 0.14 |
| ✓ | | ✓ | 67.76 ± 1.02 | 6.24 ± 0.97 | 3.53 ± 0.75 | 6.39 ± 0.69 | 68.04 ± 1.79 | 8.01 ± 1.27 | 2.58 ± 0.68 | 9.01 ± 0.68 |
| | ✓ | ✓ | 68.41 ± 1.03 | 5.99 ± 0.57 | 3.45 ± 0.43 | 6.49 ± 0.43 | 67.43 ± 0.30 | 7.37 ± 0.24 | 2.41 ± 0.21 | 9.18 ± 0.06 |
| ✓ | ✓ | ✓ | **73.33 ± 0.91** | **3.66 ± 0.64** | **2.37 ± 0.45** | **6.10 ± 0.26** | **72.40 ± 1.35** | **6.34 ± 0.68** | **2.09 ± 0.32** | **8.51 ± 0.52** |

Table 5: Performance comparison (%) with different masking methods with 128 bits on UTKFace

| Method | TA: ethnicity, SA: age | | | | TA: age, SA: ethnicity | | | |
|---|---|---|---|---|---|---|---|---|
| | MAP ↑ | EOD ↓ | EOP ↓ | DP ↓ | MAP ↑ | EOD ↓ | EOP ↓ | DP ↓ |
| No-Mask | 72.10 ± 1.43 | 4.35 ± 0.90 | 2.78 ± 0.61 | 6.45 ± 0.35 | 71.70 ± 1.62 | 6.82 ± 0.80 | 2.36 ± 0.38 | 8.74 ± 0.60 |
| Random | 71.34 ± 1.53 | 4.41 ± 0.95 | 2.83 ± 0.62 | 6.79 ± 0.46 | 70.60 ± 1.55 | 6.90 ± 0.82 | 3.01 ± 0.47 | 8.80 ± 0.62 |
| 1/K | 72.95 ± 1.10 | 3.88 ± 0.70 | 2.49 ± 0.50 | 6.22 ± 0.30 | 72.10 ± 1.20 | 6.55 ± 0.75 | 2.88 ± 0.33 | 8.60 ± 0.45 |
| DISH | **73.33 ± 0.91** | **3.66 ± 0.64** | **2.37 ± 0.45** | **6.10 ± 0.26** | **72.40 ± 1.35** | **6.34 ± 0.68** | **2.09 ± 0.32** | **8.51 ± 0.52** |

Table 6: Performance comparison (%) with different model variants with 128 bits on UTKFace

| Method | TA: ethnicity, SA: age | | | | TA: age, SA: ethnicity | | | |
|---|---|---|---|---|---|---|---|---|
| | MAP ↑ | EOD ↓ | EOP ↓ | DP ↓ | MAP ↑ | EOD ↓ | EOP ↓ | DP ↓ |
| Variant 1 | 70.40 ± 2.95 | 6.35 ± 2.10 | 3.28 ± 1.60 | 6.90 ± 0.95 | 70.70 ± 2.20 | 7.25 ± 1.12 | 2.38 ± 0.55 | 8.85 ± 1.00 |
| Variant 2 | 71.10 ± 1.35 | 4.92 ± 0.75 | 2.70 ± 0.60 | 6.60 ± 0.38 | 71.50 ± 1.18 | 7.05 ± 0.72 | 2.25 ± 0.42 | 8.88 ± 0.56 |
| Variant 3 | 72.05 ± 1.12 | 4.50 ± 0.70 | 2.62 ± 0.50 | 6.35 ± 0.34 | 71.90 ± 1.22 | 6.80 ± 0.68 | 2.20 ± 0.36 | 8.65 ± 0.52 |
| Variant 4 | 70.95 ± 1.55 | 6.15 ± 0.82 | 3.18 ± 0.62 | 6.95 ± 0.42 | 71.05 ± 1.48 | 7.50 ± 0.80 | 2.45 ± 0.46 | 9.00 ± 0.60 |
| DISH | **73.33 ± 0.91** | **3.66 ± 0.64** | **2.37 ± 0.45** | **6.10 ± 0.26** | **72.40 ± 1.35** | **6.34 ± 0.68** | **2.09 ± 0.32** | **8.51 ± 0.52** |

**Effect Of Individual Loss Functions.** Table 4 shows that removing all fairness-related objectives leads to a substantial drop in both MAP and fairness metrics. Adding the disentangled consistency loss alone yields modest MAP gains and small yet consistent reductions in disparities, as it mainly promotes semantic concentration rather than directly suppressing sensitive leakage. Introducing channel adversarial loss or conditional covariance regularization independently improves fairness and also increases MAP, though the improvements are smaller and less balanced than the full model. Combined with disentangled consistency, each component contributes complementary benefits, and the full objective achieves the best overall trade-off between accuracy and fairness.

**Comparison Of Masking Strategies.** Table 5 presents a comparison between the proposed probability-driven channel masking and several alternative masking schemes. The no-mask and random baselines achieve reasonable MAP but exhibit higher group disparities, as they fail to leverage the learned factor assignment structure and therefore cannot effectively concentrate semantic evidence. A naive $1/K$ uniform weighting offers marginal improvements by balancing channel contributions, but it still neglects the semantic concentration property learned through disentanglement. In contrast, our probability-driven masking adaptively emphasizes channels with higher assignment probabilities while attenuating those with weaker semantic relevance. Importantly, this process does not rely on target labels or sensitive attributes; instead, it exploits the intrinsic factor probabilities to preserve discriminative semantics. As a result, DISH achieves both the highest MAP and the lowest group disparities, demonstrating that adaptive, data-driven channel weighting is critical to simultaneously maintaining retrieval accuracy and enhancing fairness.

**Evaluation Of Model Variants.** Finally, we investigate four structural variants of DISH, as shown in Table 6. Replacing supervised disentangled consistency learning with unsupervised contrastive learning (Variant 1) weakens semantic concentration, resulting in significantly lower MAP and unstable fairness. Substituting the channel-wise adversary with a single global adversary (Variant 2) yields better results than Variant 1 but still lags behind DISH, highlighting the importance of suppressing sensitive leakage at a finer granularity. Using a simple decorrelation instead of conditional

covariance regularization (Variant 3) provides modest improvements, but it lacks sensitivity-specific conditioning and fails to achieve optimal fairness. Constraining fairness only in the Hamming space (Variant 4) further underperforms, confirming that interventions after binarization are insufficient due to the loss of representational flexibility. By contrast, the full DISH framework consistently achieves a better trade-off between accuracy and fairness, validating the necessity of its disentangled, multi-level suppression strategy.

**Effects Of Hyperparameters.** We assess three factors: ranked samples, channel count $K$, and loss weights $(\lambda_1, \lambda_2)$. For ranked list depth, we compare DISH with FATE, DFH, and CE(Figure 2). For $K$ and $(\lambda_1, \lambda_2)$, we conduct ablations of DISH: an intermediate $K$ and moderate loss weights provide better fairness-utility trade-off (Figures 3 and 4). Full results and details are in appendix D.

## 6 DISCUSSION

We further assess the robustness and generalizability of DISH through extended studies summarized here and detailed in the AppendixF. On the DeepFashion dataset, which exhibits richer semantics and stronger appearance shifts than facial benchmarks, DISH consistently surpasses existing hashing methods in both retrieval quality and fairness (Appendix F.1). We also compare against state-of-the-art fair representation learning methods (VAE-based and adversarial) under joint end-to-end and two-stage training; simply attaching a hashing head to these baselines yields inferior fairness–accuracy trade-offs, highlighting the need to jointly optimize disentanglement, fairness, and quantization (Appendix F.2). Finally, experiments with multi-class sensitive attributes (e.g., four-way age groups) and intersectional attributes (ethnicity × gender) show that DISH maintains superior performance without architectural changes, indicating robust adaptation to complex and fine-grained sensitive structures (Appendix F.3). Together, these results demonstrate that DISH remains effective across datasets, training protocols, and sensitive-attribute configurations, supporting its practical applicability in real-world fair retrieval scenarios.

## 7 CONCLUSION

We introduce DISH, a fairness-aware deep hashing framework that combines disentangled consistency, probability-driven masking, channel-wise adversaries, and conditional covariance regularization. Extensive experiments on multiple benchmarks demonstrate that DISH consistently outperforms existing methods, achieving state-of-the-art retrieval accuracy together with substantially improved fairness. The ablation studies further validate the necessity of each component and confirm that their integration yields a clear Pareto improvement.

## 8 ETHICS STATEMENT

This research addresses a critical ethical concern in large-scale image retrieval systems: the potential for algorithmic bias that may lead to discriminatory outcomes across demographic groups defined by sensitive attributes such as age, gender, and ethnicity. The development and deployment of biased retrieval systems can perpetuate societal inequalities, reinforce stereotypes, and cause harm to marginalized communities through unequal access to information and opportunities. In this work, we explicitly acknowledge the ethical implications of biased representation learning and commit to developing solutions that promote fairness without compromising utility. Our dataset selection and annotation processes were conducted with careful consideration of demographic representation, ensuring balanced coverage across sensitive attribute categories where possible. We strictly adhered to the ethical guidelines established by the original dataset creators (UTKFace and CelebA), which include informed consent protocols for image collection and usage. Importantly, our framework DISH actively works to disentangle and suppress sensitive information while preserving semantic utility, thereby reducing the risk of discriminatory retrieval results. We recognize that fairness is a complex, context-dependent concept, and our approach focuses on group fairness metrics (demographic parity, equal opportunity, and equalized odds) as measurable objectives. While our method significantly improves fairness outcomes, we acknowledge that no technical solution can fully resolve deeply rooted societal biases, and we advocate for continued interdisciplinary collaboration between computer scientists, social scientists, and policymakers to address these challenges holistically.

## 9 REPRODUCIBILITY STATEMENT

To ensure the reproducibility of our research findings, we have implemented rigorous experimental protocols and documentation practices throughout this study. All experiments were conducted using publicly available datasets (UTKFace and CelebA) with clearly specified preprocessing procedures, including detailed descriptions of how target and sensitive attributes were defined and binarized for each experimental configuration. For each experiment, we report results averaged over five independent runs with different random seeds, presenting both mean values and standard deviations to demonstrate statistical reliability. Our complete experimental setup including network architectures (ResNet-50 backbone with specified modifications), optimization parameters (learning rates, batch sizes, optimizer configurations), and hyperparameter settings ($\lambda_1$, $\lambda_2$ are explicitly documented in the methodology section and supplementary materials. The ablation studies systematically evaluate the contribution of each component in our framework, providing comprehensive evidence for our design choices. To facilitate comparison with existing methods, we have reimplemented all baseline approaches following their original publications and verified their performance against reported results where possible. We welcome replication attempts and are committed to providing support to researchers seeking to reproduce or build upon our work.

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

## A   THE USE OF LARGE LANGUAGE MODELS

We used a large language model–based assistant solely for linguistic polishing (e.g., grammar, style, clarity, and minor LaTeX formatting). All technical content, including problem formulation, algorithms, analyses, and conclusions, was conceived and verified by the authors. All references were selected and checked by the authors. Outputs from the assistant were reviewed and edited for accuracy; any remaining errors are the authors' responsibility.

## B   ALGORITHM PROCEDURE

---

**Algorithm 1** Algorithm procedure of DISH

---

**Input**: Dataset $\mathcal{D} = \{I_i, y_i, s_i\}_{i=1}^N$, code length $L$, channel number $K$, weights $(\lambda_1, \lambda_2)$

**Output**: Encoder & prototypes $\theta$ (incl. $\{\mathbf{c}_k\}_{k=1}^K$), hash head $\eta$, channel discriminators $\Psi = \{\psi_k\}_{k=1}^K$, semantic anchors $\{\mathbf{z}_c\}_{c=1}^C$

1: **Init**: Randomly initialize $\theta$ (encoder and prototypes $\{\mathbf{c}_k\}$), hash head $f_\eta$, and discriminators $\{D_{\psi_k}\}_{k=1}^K$; set anchors $\{\mathbf{z}_c\}$ by Rademacher.

2: **Main training**

3: **for** $e = 1$ **to** $E$ **do**

4:     **for** minibatches $\mathcal{B} \subset \mathcal{D}$ **do**

5:       *Augment*: For each $I_i \in \mathcal{B}$, sample $I_i'$.

6:       *Encode*: $\mathbf{x}_{i,1:K}$, $\mathbf{x}_{i,1:K}' \leftarrow$ encoder $\theta$.

7:       *Factor assignments*: $p_\theta(k\,|\,\mathbf{x}_i)$ by Eq. equation 1.

8:       *DCL*: compute per-factor $\ell_{i,k}$ and $q_\theta(k\,|\,\mathbf{x}_i, y_i)$ by Eqs. 7–6; get $\mathcal{L}_{\mathrm{DCL}}$ by Eq. 11.

9:       *Masking*: $\mathbf{m}_i = [p_\theta(k\,|\,\mathbf{x}_i)]_{k=1}^K$ and $\tilde{\mathbf{x}}_i = \mathbf{x}_i \odot_c \mathbf{m}_i$ by Eqs. 12–13.

10:      *Channel adversary*: for each $k$, predict $D_{\psi_k}(\tilde{\mathbf{x}}_{i,k})$ and compute $\mathcal{L}_{\mathrm{CAL}}$ by Eq. 14.

11:      *Conditional covariance*: compute $\mathcal{L}_{\mathrm{CCR}}$ on $\{\tilde{\mathbf{x}}_{i,k}\}$ grouped by $s_i$ via Eq. 17.

12:      *Semantic alignment*: $\mathbf{u}_i = f_\eta(\tilde{\mathbf{x}}_i)$; compute $\mathcal{L}_{\mathrm{SA}}$ by Eq. 21.

13:      *Joint objective (saddle point)*: $\mathcal{L} = \mathcal{L}_{\mathrm{DCL}} - \lambda_1 \mathcal{L}_{\mathrm{CAL}} + \lambda_2 \mathcal{L}_{\mathrm{CCR}} + \mathcal{L}_{\mathrm{SA}}$ (Eq. 22)

14:      **Update** $\Psi$: minimize $\mathcal{L}_{\mathrm{CAL}}$ w.r.t. $\{\psi_k\}$ (standard gradient).

15:      **Update** $\theta, \eta, \{\mathbf{z}_c\}$: minimize $\mathcal{L}$ (GRL implements the "$-\lambda_1 \mathcal{L}_{\mathrm{CAL}}$" effect on $\theta$).

16:     **end for**

17: **end for**

18: **Inference**: given $I$, compute $\tilde{\mathbf{x}}$ and $\mathbf{u} = f_\eta(\tilde{\mathbf{x}})$; output $\mathbf{b} = \mathrm{sign}(\mathbf{u})$.

---

## C   BASELINES

To validate the effectiveness of our method, we conduct comprehensive comparisons with state-of-the-art deep hashing approaches. The selected baselines include: OrthoHash (Hoe et al., 2021), Bihalf (Li & van Gemert, 2021), CE (Li et al., 2017), CSQ (Yuan et al., 2020), DFH (Li et al., 2019), DPSH (Li et al., 2015) , DTSH (Wang et al., 2017) , GreedyHash (Su et al., 2018), SDH-C (Shen et al., 2015) , DLBD (Xiao et al., 2023), MDSHC (Wang et al., 2023) and FATE (Zhang et al., 2024).

## D   PARAMETER SENSITIVITY ANALYSIS

We study the sensitivity of DISH to three factors: the number of ranked samples, the channel count $K$, and the loss weights $\lambda_1$ (channel adversary) and $\lambda_2$ (conditional covariance). Varying the list depth shows that DISH sustains its fairness advantages over competitive baselines (FATE, DFH, CE) across a wide range of ranked samples without compromising retrieval accuracy, and the relative ordering among methods remains stable (Figure 2). Adjusting $K$ yields a robust U-shaped trend (Figure 3): too few channels under-disentangle factors, while too many fragment semantics and weaken masking/regularization; an intermediate $K$ delivers the best fairness–utility balance across code lengths. The $(\lambda_1, \lambda_2)$ landscape (Figure 4) shows a similar concavity: a moderate-to-strong $\lambda_1$ paired with a moderate $\lambda_2$ forms a stable ridge that maximizes the Pareto frontier. As further

visualized in Figure 4, both UTKFace and CelebA exhibit a broad region of stability, where setting $\lambda_1$ and $\lambda_2$ within the range $[0.01, 0.1]$ consistently yields good fairness performance.

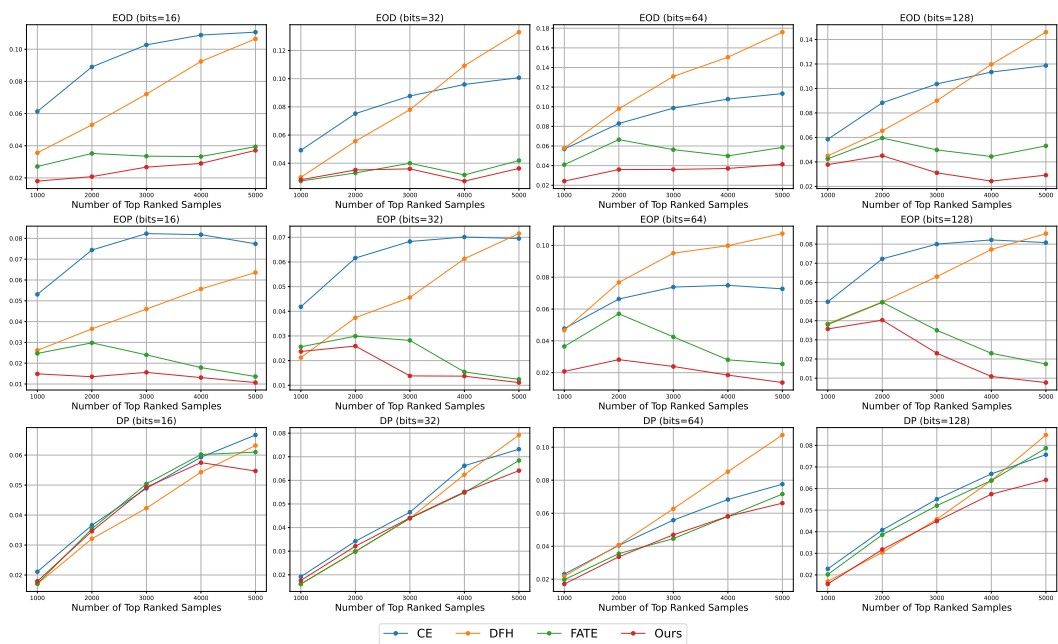

Figure 2: Sensitivity analysis of ranked samples with code lengths 128 on UTKFace. Target Attribute: age , Sensitive Attribute: ethnicity.

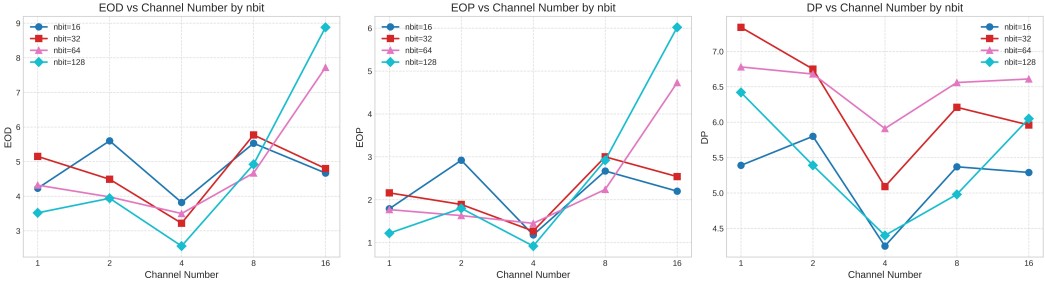

Figure 3: Sensitivity analysis of channel number $K$ with code lengths 128 on UTKFace. Target Attribute: ethnicity, Sensitive Attribute: age.

# E IMPLEMENTATION DETAILS

All experiments were conducted on a single NVIDIA RTX 4090 GPU (24 GB) and, unless otherwise noted, each configuration was trained for 100 epochs. To ensure strict comparability, DISH and all baselines use the same ResNet-50 backbone with identical input resolution, normalization, and augmentation; only method-specific heads and loss terms differ. Data splits and the query/gallery partitions are fixed and reused across methods. For each dataset and configuration, we run five independent trials and report mean $\pm$ std. Within each dataset, all methods share the same training recipe (batch size, optimizer, learning-rate schedule, weight decay, and augmentation pipeline). We control randomness with an identical set of random seeds for every method and trial, applied consistently to Python, NumPy, and PyTorch (including dataloader workers and CUDA determinism);

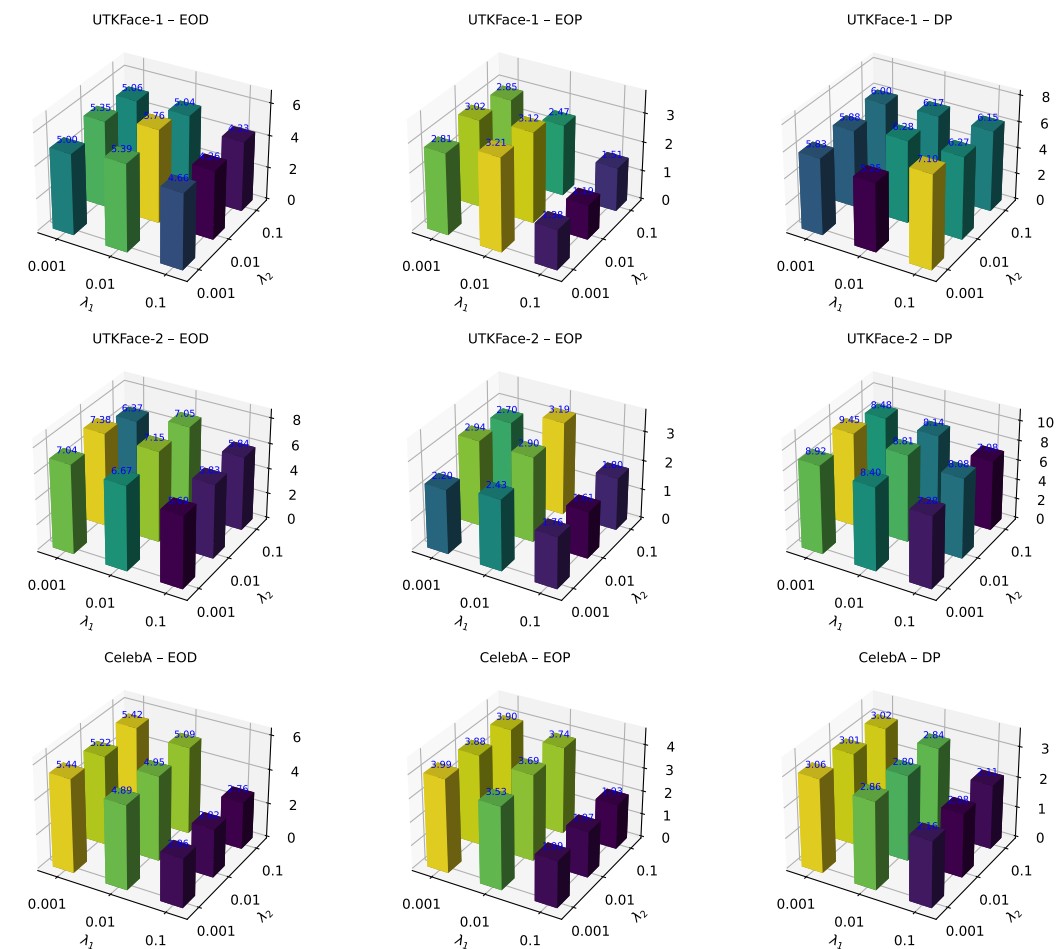

Figure 4: Visualization of the hyper-parameter sweep over the adversarial ($\lambda_1$) and covariance ($\lambda_2$) regularization weights for **DISH** under code lengths 128. Each subplot shows a 3D bar plot of a fairness metric (columns: EOD, EOP, DP) across different ($\lambda_1, \lambda_2$) combinations for three datasets (rows): **UTKFace-1** (*Target: ethnicity, Sensitive: age*), **UTKFace-2** (*Target: age, Sensitive: ethnicity*), and **CelebA** (*Target: attractiveness, Sensitive: male*).

# F ADDITIONAL DISCUSSION

## F.1 EXPERIMENTS ON NATURAL IMAGE DATASETS

To further evaluate the robustness and generalizability of our framework beyond facial imagery, we conduct additional experiments on the DeepFashion dataset(Liu et al., 2016)—a large-scale real-world clothing retrieval benchmark containing over 200,000 images, more than 1,000 fine-grained attributes, and 50+ clothing categories. Compared with UTKFace, DeepFashion exhibits substantially higher intra-class variation, stronger appearance shifts, and richer semantic diversity, making it a challenging testbed for both retrieval and fairness. Following the standard fairness-aware setting, we treat *attribute* prediction as the target task and *clothing category* as the sensitive attribute. Across all four code lengths (16–128 bits), DISH consistently achieves the best performance in both retrieval quality (MAP) and fairness metrics (EOD/EOP/DP), confirming its effectiveness on complex, non-face natural image domains.

Table 7: Performance comparison (%) with the state-of-the-art methods on DeepFashion with code lengths varying from 16 to 128. Target Attribute: attribute, Sensitive Attribute: category.

| Method | 16 bits | | | | 32 bits | | | | 64 bits | | | | 128 bits | | | |
|---|---|---|---|---|---|---|---|---|---|---|---|---|---|---|---|---|
| | MAP ↑ | EOD ↓ | EOP ↓ | DP ↓ | MAP ↑ | EOD ↓ | EOP ↓ | DP ↓ | MAP ↑ | EOD ↓ | EOP ↓ | DP ↓ | MAP ↑ | EOD ↓ | EOP ↓ | DP ↓ |
| OrthoHash | 28.09 ± 0.47 | 9.74 ± 0.04 | 5.79 ± 0.21 | 4.03 ± 0.18 | 29.71 ± 0.07 | 6.20 ± 0.36 | 3.91 ± 0.31 | 2.26 ± 0.17 | 30.00 ± 0.81 | 6.77 ± 0.29 | 4.07 ± 0.25 | 2.66 ± 0.16 | 31.53 ± 0.59 | 7.26 ± 1.21 | 4.86 ± 0.88 | 2.43 ± 0.36 |
| Bihalf | 33.52 ± 0.45 | 25.11 ± 0.98 | 14.98 ± 0.71 | 10.80 ± 0.10 | 30.08 ± 0.21 | 23.10 ± 0.57 | 13.14 ± 0.50 | 10.53 ± 0.18 | 27.35 ± 0.34 | 17.65 ± 0.20 | 10.64 ± 0.13 | 7.48 ± 0.13 | 26.97 ± 0.70 | 22.80 ± 0.26 | 13.22 ± 0.16 | 10.05 ± 0.14 |
| CE | 28.53 ± 0.38 | 8.96 ± 0.61 | 4.98 ± 0.52 | 4.05 ± 0.11 | 29.21 ± 0.88 | 7.56 ± 0.71 | 4.44 ± 0.51 | 3.09 ± 0.30 | 29.18 ± 0.14 | 9.10 ± 0.32 | 5.26 ± 0.25 | 3.86 ± 0.10 | 30.05 ± 0.52 | 9.46 ± 0.12 | 5.40 ± 0.16 | 4.13 ± 0.11 |
| CSQ | 30.42 ± 0.94 | 7.08 ± 0.45 | 4.21 ± 0.32 | 3.01 ± 0.16 | 32.33 ± 1.43 | 17.09 ± 0.49 | 10.20 ± 0.32 | 7.21 ± 0.21 | 33.48 ± 0.21 | 13.50 ± 0.41 | 8.42 ± 0.33 | 5.37 ± 0.15 | 32.44 ± 2.10 | 17.04 ± 0.35 | 9.82 ± 0.19 | 7.43 ± 0.15 |
| DFH | 37.47 ± 0.82 | 7.65 ± 0.10 | 5.42 ± 0.10 | 2.55 ± 0.04 | 33.64 ± 0.67 | 11.27 ± 0.10 | 7.19 ± 0.18 | 4.45 ± 0.10 | 32.70 ± 0.96 | 15.27 ± 0.22 | 9.58 ± 0.29 | 6.21 ± 0.20 | 31.41 ± 0.17 | 13.24 ± 0.17 | 8.62 ± 0.17 | 5.06 ± 0.11 |
| DPSH | 32.17 ± 0.88 | 7.65 ± 1.10 | 4.65 ± 0.74 | 3.07 ± 0.61 | 32.02 ± 0.80 | 7.11 ± 0.89 | 4.20 ± 0.69 | 2.90 ± 0.14 | 32.08 ± 0.05 | 6.35 ± 1.72 | 3.83 ± 1.12 | 2.47 ± 0.63 | 32.71 ± 2.60 | 5.64 ± 1.40 | 3.46 ± 1.30 | 2.15 ± 0.67 |
| DTSH | 30.97 ± 0.47 | 5.98 ± 0.11 | 3.46 ± 0.19 | 2.47 ± 0.04 | 34.23 ± 0.79 | 9.02 ± 0.46 | 5.43 ± 0.36 | 3.85 ± 0.14 | 30.47 ± 0.96 | 9.80 ± 0.54 | 6.47 ± 0.52 | 3.57 ± 0.28 | 30.02 ± 1.80 | 8.30 ± 0.20 | 5.71 ± 0.01 | 2.83 ± 0.05 |
| GreedyHash | 31.19 ± 0.82 | 11.17 ± 0.28 | 6.99 ± 0.10 | 4.51 ± 0.13 | 29.67 ± 1.12 | 7.12 ± 0.23 | 4.24 ± 0.10 | 2.88 ± 0.11 | 30.06 ± 1.08 | 7.62 ± 0.58 | 4.33 ± 0.33 | 3.27 ± 0.14 | 29.21 ± 0.70 | 9.92 ± 0.39 | 5.40 ± 0.26 | 4.60 ± 0.05 |
| SDH-C | 29.83 ± 0.73 | 11.30 ± 0.17 | 6.82 ± 0.16 | 4.55 ± 0.15 | 29.43 ± 0.14 | 13.76 ± 0.22 | 8.14 ± 0.20 | 5.78 ± 0.29 | 29.34 ± 0.24 | 14.65 ± 0.10 | 8.63 ± 0.11 | 6.38 ± 0.13 | 27.77 ± 0.74 | 17.49 ± 0.36 | 9.91 ± 0.21 | 7.93 ± 0.16 |
| DLBD | 23.88 ± 0.03 | 14.67 ± 0.10 | 6.67 ± 0.05 | 7.81 ± 0.07 | 24.14 ± 0.02 | 13.47 ± 0.12 | 6.21 ± 0.06 | 7.16 ± 0.06 | 24.38 ± 0.02 | 12.90 ± 0.08 | 5.88 ± 0.04 | 6.88 ± 0.04 | 24.51 ± 0.01 | 13.73 ± 0.05 | 6.25 ± 0.02 | 7.37 ± 0.03 |
| MDSHC | 27.79 ± 0.08 | 8.22 ± 0.18 | 3.97 ± 0.10 | 4.16 ± 0.10 | 27.01 ± 0.10 | 8.37 ± 0.20 | 4.50 ± 0.08 | 3.80 ± 0.13 | 26.48 ± 0.08 | 14.24 ± 0.16 | 7.21 ± 0.09 | 7.03 ± 0.08 | 24.55 ± 0.05 | 14.50 ± 0.12 | 6.81 ± 0.05 | 7.60 ± 0.08 |
| FATE | 41.46 ± 0.46 | 3.90 ± 0.08 | 2.70 ± 0.06 | 2.50 ± 0.05 | 42.61 ± 0.52 | 4.60 ± 0.13 | 2.40 ± 0.07 | 2.10 ± 0.09 | 42.89 ± 0.36 | 3.93 ± 0.06 | 1.96 ± 0.05 | 1.90 ± 0.04 | 42.35 ± 0.38 | 3.50 ± 0.11 | 1.90 ± 0.05 | 1.70 ± 0.08 |
| **DISH** | **47.13 ± 0.35** | **2.96 ± 0.33** | **1.65 ± 0.18** | **1.49 ± 0.18** | **48.53 ± 0.45** | **2.86 ± 0.22** | **1.83 ± 0.10** | **1.77 ± 0.14** | **48.40 ± 0.44** | **2.74 ± 0.10** | **1.81 ± 0.08** | **1.87 ± 0.05** | **48.98 ± 0.83** | **2.45 ± 0.35** | **1.84 ± 0.19** | **1.63 ± 0.16** |

## F.2 COMPARISON WITH FAIR REPRESENTATION BASELINES

To rigorously validate the necessity of our proposed architecture, we compare DISH against representative state-of-the-art fair representation learning methods, including VAE-based approaches ($\beta$-VAE(Higgins et al., 2017), FactorVAE(Kim & Mnih, 2018), FFVAE(Creager et al., 2019)) and adversarial/disentanglement-based approaches (AFR(Zhang et al., 2018), SADZhu et al. (2024), CDFG(Zhang et al., 2025a)). To ensure a fair comparison, we evaluate these baselines under two distinct protocols to determine whether existing fair continuous representations can be trivially adapted for fair hashing. **Protocol 1: Joint End-to-End Training.** In this setting, we attach the same hashing head and semantic alignment loss used in DISH to the baselines and train the entire network jointly. As shown in Table 8 and Table 9, simply augmenting existing methods with a hashing objective yields suboptimal results. VAE-based methods ($\beta$-VAE, FactorVAE, FFVAE) suffer from poor retrieval accuracy (MAP ≈ 20–32%) and large fairness gaps, indicating that their reconstruction-oriented latent spaces lack the discriminative power required for effective hashing. While other baselines (AFR, SAD, CDFG) achieve reasonable MAP, they consistently underperform DISH in both accuracy and fairness metrics (EOD/EOP/DP). **Protocol 2: Two-Stage Training.** We further investigate whether a decoupled strategy—learning a fair continuous representation first, then freezing it to train a hashing module—is effective. The results in Table 10 and Table 11 reveal a consistent performance degradation compared to the joint training setting. For all methods, the two-stage approach leads to strictly lower MAP and inflated fairness metrics. This empirical evidence underscores that fairness and semantic discriminability should be enforced jointly with the quantization objective.

Table 8: Comparison of representative fair-representation learning approaches on UTKFace under **joint end-to-end training** with code lengths varying from 16 to 128. Target Attribute: age, Sensitive Attribute: ethnicity.

| Method | 16 bits | | | | 32 bits | | | | 64 bits | | | | 128 bits | | | |
|---|---|---|---|---|---|---|---|---|---|---|---|---|---|---|---|---|
| | MAP ↑ | EOD ↓ | EOP ↓ | DP ↓ | MAP ↑ | EOD ↓ | EOP ↓ | DP ↓ | MAP ↑ | EOD ↓ | EOP ↓ | DP ↓ | MAP ↑ | EOD ↓ | EOP ↓ | DP ↓ |
| $\beta$-VAE | 28.20 ± 0.24 | 11.36 ± 0.23 | 6.56 ± 0.18 | 5.45 ± 0.09 | 31.69 ± 0.17 | 14.53 ± 0.13 | 8.30 ± 0.08 | 7.29 ± 0.10 | 30.52 ± 0.09 | 14.83 ± 0.16 | 8.04 ± 0.10 | 7.35 ± 0.06 | 30.92 ± 0.05 | 14.61 ± 0.09 | 8.22 ± 0.06 | 6.85 ± 0.06 |
| FactorVAE | 31.68 ± 1.17 | 8.93 ± 0.31 | 5.03 ± 0.15 | 5.12 ± 0.20 | 25.12 ± 0.58 | 9.12 ± 0.43 | 3.49 ± 0.51 | 7.85 ± 0.64 | 25.56 ± 0.48 | 8.64 ± 0.21 | 4.22 ± 0.09 | 6.31 ± 0.31 | 27.16 ± 0.92 | 10.08 ± 0.62 | 5.50 ± 0.56 | 6.34 ± 0.33 |
| FFVAE | 25.13 ± 0.08 | 9.18 ± 0.07 | 4.66 ± 0.32 | 9.92 ± 0.10 | 24.68 ± 0.03 | 9.49 ± 0.05 | 4.49 ± 0.10 | 10.38 ± 0.08 | 24.62 ± 0.03 | 9.26 ± 0.05 | 4.44 ± 0.07 | 10.21 ± 0.10 | 25.52 ± 0.35 | 9.02 ± 0.15 | 4.53 ± 0.18 | 9.88 ± 0.12 |
| AFR | 63.33 ± 0.80 | 9.82 ± 0.36 | 6.44 ± 0.33 | 6.69 ± 0.33 | 67.29 ± 0.70 | 10.50 ± 0.62 | 6.65 ± 0.52 | 7.33 ± 0.26 | 62.22 ± 1.22 | 9.02 ± 0.28 | 5.23 ± 0.26 | 6.27 ± 0.28 | 66.31 ± 0.79 | 13.20 ± 0.50 | 8.94 ± 0.45 | 7.62 ± 0.37 |
| SAD | 55.58 ± 0.96 | 8.35 ± 0.44 | 4.65 ± 0.29 | 6.55 ± 0.17 | 61.97 ± 0.62 | 6.57 ± 0.26 | 3.36 ± 0.17 | 7.07 ± 0.14 | 63.21 ± 1.11 | 7.47 ± 0.24 | 4.38 ± 0.24 | 5.93 ± 0.20 | 58.70 ± 0.70 | 8.98 ± 0.36 | 4.72 ± 0.27 | 6.99 ± 0.16 |
| CDFG | 56.94 ± 0.84 | 6.53 ± 0.37 | 3.58 ± 0.24 | 5.23 ± 0.23 | 57.33 ± 0.75 | 10.14 ± 0.31 | 5.29 ± 0.16 | 7.85 ± 0.16 | 65.86 ± 0.72 | 6.85 ± 0.20 | 3.97 ± 0.15 | 6.22 ± 0.14 | 62.43 ± 0.31 | 8.65 ± 0.36 | 5.22 ± 0.28 | 6.22 ± 0.18 |
| **DISH** | **72.99 ± 1.32** | **4.22 ± 0.75** | **2.08 ± 0.75** | **4.79 ± 0.15** | **73.20 ± 1.64** | **4.12 ± 0.57** | **2.55 ± 0.49** | **6.54 ± 0.28** | **73.67 ± 1.83** | **4.28 ± 0.98** | **2.96 ± 0.84** | **5.63 ± 0.37** | **73.33 ± 0.91** | **3.66 ± 0.64** | **2.37 ± 0.45** | **6.10 ± 0.26** |

Table 9: Comparison of representative fair-representation learning approaches on UTKFace under **joint end-to-end training** with code lengths varying from 16 to 128. Target Attribute: ethnicity, Sensitive Attribute: age.

| Method | 16 bits | | | | 32 bits | | | | 64 bits | | | | 128 bits | | | |
|---|---|---|---|---|---|---|---|---|---|---|---|---|---|---|---|---|
| | MAP ↑ | EOD ↓ | EOP ↓ | DP ↓ | MAP ↑ | EOD ↓ | EOP ↓ | DP ↓ | MAP ↑ | EOD ↓ | EOP ↓ | DP ↓ | MAP ↑ | EOD ↓ | EOP ↓ | DP ↓ |
| $\beta$-VAE | 31.42 ± 0.13 | 11.53 ± 0.14 | 6.71 ± 0.08 | 10.38 ± 0.12 | 31.59 ± 0.10 | 21.31 ± 0.19 | 11.05 ± 0.16 | 11.02 ± 0.06 | 32.51 ± 0.10 | 15.15 ± 0.31 | 9.21 ± 0.18 | 9.83 ± 0.18 | 31.34 ± 0.06 | 20.47 ± 0.16 | 11.12 ± 0.06 | 9.99 ± 0.11 |
| FactorVAE | 24.07 ± 0.50 | 16.69 ± 0.48 | 6.81 ± 0.50 | 10.81 ± 0.11 | 21.74 ± 0.59 | 13.54 ± 0.34 | 4.48 ± 0.16 | 10.48 ± 0.21 | 21.68 ± 0.59 | 9.45 ± 1.19 | 3.59 ± 1.17 | 9.79 ± 0.33 | 24.34 ± 0.36 | 19.03 ± 1.31 | 9.82 ± 1.37 | 9.77 ± 0.26 |
| FFVAE | 23.12 ± 0.42 | 14.85 ± 0.36 | 5.78 ± 0.41 | 9.92 ± 0.28 | 22.05 ± 0.47 | 12.61 ± 0.31 | 4.02 ± 0.21 | 9.88 ± 0.33 | 21.94 ± 0.55 | 9.12 ± 0.97 | 3.01 ± 0.81 | 9.24 ± 0.45 | 22.86 ± 0.63 | 17.25 ± 1.02 | 8.96 ± 1.11 | 9.11 ± 0.58 |
| AFR | 57.67 ± 0.48 | 12.28 ± 0.63 | 6.78 ± 0.46 | 8.21 ± 0.29 | 59.05 ± 0.54 | 12.38 ± 0.29 | 6.22 ± 0.21 | 9.33 ± 0.15 | 57.21 ± 0.76 | 14.33 ± 0.40 | 6.92 ± 0.31 | 10.38 ± 0.18 | 60.64 ± 0.69 | 14.94 ± 0.33 | 6.86 ± 0.23 | 11.69 ± 0.19 |
| SAD | 51.11 ± 0.57 | 12.99 ± 0.41 | 5.59 ± 0.22 | 9.81 ± 0.32 | 58.46 ± 0.49 | 12.79 ± 0.33 | 4.91 ± 0.21 | 11.22 ± 0.21 | 56.26 ± 0.93 | 13.85 ± 0.28 | 6.43 ± 0.14 | 10.71 ± 0.23 | 56.99 ± 0.71 | 10.37 ± 0.20 | 3.89 ± 0.14 | 9.23 ± 0.16 |
| CDFG | 51.91 ± 0.63 | 12.68 ± 0.32 | 5.50 ± 0.28 | 9.49 ± 0.22 | 53.41 ± 0.68 | 12.13 ± 0.40 | 4.22 ± 0.19 | 10.80 ± 0.38 | 55.88 ± 0.83 | 11.89 ± 0.19 | 4.67 ± 0.13 | 9.85 ± 0.13 | 57.93 ± 0.40 | 10.47 ± 0.34 | 4.55 ± 0.22 | 8.76 ± 0.22 |
| **DISH** | **72.08 ± 0.71** | **6.46 ± 0.49** | **1.41 ± 0.38** | **7.08 ± 0.39** | **71.85 ± 1.82** | **6.76 ± 0.24** | **1.59 ± 0.22** | **8.14 ± 0.26** | **71.59 ± 0.51** | **6.03 ± 0.35** | **1.29 ± 0.20** | **8.83 ± 0.38** | **72.40 ± 1.35** | **6.34 ± 0.68** | **2.09 ± 0.32** | **8.51 ± 0.52** |

Table 10: Comparison of fair-representation approaches under a **two-stage training pipeline**. In Stage 1, each method learns a fair representation **without** any hashing loss; in Stage 2, a separate hashing module is trained on top of the learned representation on UTKFace with code lengths varying from 16 to 128. Target Attribute: ethnicity, Sensitive Attribute: age.

| Method | 16 bits | | | | 32 bits | | | | 64 bits | | | | 128 bits | | | |
|---|---|---|---|---|---|---|---|---|---|---|---|---|---|---|---|---|
| | MAP ↑ | EOD ↓ | EOP ↓ | DP ↓ | MAP ↑ | EOD ↓ | EOP ↓ | DP ↓ | MAP ↑ | EOD ↓ | EOP ↓ | DP ↓ | MAP ↑ | EOD ↓ | EOP ↓ | DP ↓ |
| $\beta$-VAE | 29.18 ± 0.34 | 15.47 ± 0.46 | 9.54 ± 0.39 | 7.63 ± 0.25 | 32.14 ± 0.29 | 18.72 ± 0.34 | 11.48 ± 0.33 | 9.57 ± 0.30 | 31.03 ± 0.27 | 18.82 ± 0.32 | 11.05 ± 0.29 | 9.36 ± 0.25 | 31.31 ± 0.26 | 18.64 ± 0.41 | 11.43 ± 0.35 | 9.05 ± 0.28 |
| FactorVAE | 25.63 ± 1.18 | 12.58 ± 0.58 | 7.46 ± 0.51 | 7.85 ± 0.35 | 24.74 ± 0.66 | 13.06 ± 0.48 | **5.25 ± 0.41** | 9.96 ± 0.38 | 25.33 ± 0.61 | 12.04 ± 0.52 | 6.16 ± 0.36 | 8.47 ± 0.33 | 27.71 ± 0.98 | 14.73 ± 0.76 | 7.95 ± 0.67 | 8.67 ± 0.41 |
| FFVAE | 25.47 ± 0.41 | 13.25 ± 0.52 | 7.66 ± 0.47 | 12.57 ± 0.44 | 25.16 ± 0.36 | 13.46 ± 0.47 | 7.18 ± 0.37 | 12.86 ± 0.42 | 24.98 ± 0.46 | 12.84 ± 0.45 | 6.86 ± 0.33 | 12.38 ± 0.43 | 25.82 ± 0.51 | 13.06 ± 0.48 | 7.32 ± 0.40 | 12.05 ± 0.46 |
| AFR | 63.82 ± 0.92 | 13.27 ± 0.67 | 8.95 ± 0.57 | 9.36 ± 0.45 | 67.72 ± 0.82 | 15.03 ± 0.72 | 9.47 ± 0.61 | 10.14 ± 0.52 | 62.81 ± 1.38 | 13.16 ± 0.64 | 7.95 ± 0.54 | 9.04 ± 0.49 | 66.92 ± 0.78 | 17.33 ± 0.79 | 11.58 ± 0.68 | 10.47 ± 0.40 |
| SAD | 56.04 ± 0.98 | 11.46 ± 0.57 | 6.63 ± 0.40 | 8.74 ± 0.35 | 62.31 ± 0.77 | **10.17 ± 0.51** | 5.47 ± 0.34 | 9.33 ± 0.33 | 63.92 ± 1.23 | 10.74 ± 0.46 | 6.27 ± 0.32 | 8.05 ± 0.35 | 59.14 ± 0.86 | 11.82 ± 0.45 | 6.43 ± 0.34 | 9.16 ± 0.32 |
| CDFG | 57.46 ± 0.90 | **10.27 ± 0.47** | 5.94 ± 0.36 | 7.65 ± 0.33 | 57.84 ± 0.69 | 13.72 ± 0.53 | 7.33 ± 0.42 | 10.25 ± 0.40 | 66.22 ± 0.85 | 10.44 ± 0.38 | 6.26 ± 0.30 | 8.53 ± 0.32 | 62.97 ± 0.57 | 12.47 ± 0.45 | 7.53 ± 0.35 | **8.65 ± 0.31** |
| **DISH** | **70.71 ± 0.63** | 13.69 ± 0.36 | **5.58 ± 0.22** | **6.31 ± 0.34** | **70.95 ± 0.72** | 10.65 ± 0.15 | 6.72 ± 0.03 | **8.35 ± 0.07** | **73.44 ± 0.46** | **9.04 ± 0.56** | **5.78 ± 0.50** | **7.35 ± 0.17** | **71.99 ± 0.36** | **10.45 ± 0.10** | **6.23 ± 0.03** | 8.98 ± 0.11 |

Table 11: Comparison of fair-representation approaches under a **two-stage training pipeline**. In Stage 1, each method learns a fair representation **without** any hashing loss; in Stage 2, a separate hashing module is trained on top of the learned representation on UTKFace with code lengths varying from 16 to 128. Target Attribute: age, Sensitive Attribute: ethnicity.

| Method | 16 bits | | | | 32 bits | | | | 64 bits | | | | 128 bits | | | |
|---|---|---|---|---|---|---|---|---|---|---|---|---|---|---|---|---|
| | MAP ↑ | EOD ↓ | EOP ↓ | DP ↓ | MAP ↑ | EOD ↓ | EOP ↓ | DP ↓ | MAP ↑ | EOD ↓ | EOP ↓ | DP ↓ | MAP ↑ | EOD ↓ | EOP ↓ | DP ↓ |
| $\beta$-VAE | 32.02 ± 0.13 | 13.53 ± 0.14 | 7.71 ± 0.08 | 11.18 ± 0.12 | 34.39 ± 0.10 | 22.81 ± 0.19 | 11.85 ± 0.16 | 12.02 ± 0.06 | 33.81 ± 0.10 | 16.15 ± 0.31 | 9.91 ± 0.18 | 10.73 ± 0.18 | 34.94 ± 0.06 | 22.17 ± 0.16 | 12.32 ± 0.06 | 11.09 ± 0.11 |
| FactorVAE | 24.67 ± 0.50 | 18.69 ± 0.48 | 7.81 ± 0.50 | 11.61 ± 0.11 | 22.54 ± 0.59 | 15.04 ± 0.34 | 5.28 ± 0.16 | 11.48 ± 0.21 | 21.98 ± 0.59 | 10.45 ± 1.19 | 4.29 ± 1.17 | 10.69 ± 0.33 | 21.24 ± 0.36 | 20.73 ± 1.31 | 11.02 ± 1.37 | 10.87 ± 0.26 |
| FFVAE | 21.72 ± 0.42 | 16.85 ± 0.36 | 6.78 ± 0.41 | 10.72 ± 0.28 | 19.85 ± 0.47 | 14.11 ± 0.31 | **4.82 ± 0.21** | 10.88 ± 0.33 | 21.24 ± 0.55 | 13.52 ± 0.60 | 5.76 ± 0.55 | 11.26 ± 0.04 | 20.46 ± 0.63 | 18.95 ± 1.02 | 10.16 ± 1.11 | 10.21 ± 0.58 |
| AFR | 58.27 ± 0.48 | 14.28 ± 0.63 | 7.78 ± 0.46 | 9.01 ± 0.29 | 59.85 ± 0.54 | 13.88 ± 0.29 | 7.02 ± 0.21 | 10.33 ± 0.15 | 57.51 ± 0.76 | 15.33 ± 0.40 | 7.62 ± 0.31 | 11.28 ± 0.18 | 60.24 ± 0.69 | 16.64 ± 0.33 | 8.06 ± 0.23 | 12.79 ± 0.19 |
| SAD | 51.71 ± 0.57 | 14.99 ± 0.41 | 6.59 ± 0.22 | 10.61 ± 0.32 | 59.26 ± 0.49 | 14.29 ± 0.33 | 5.71 ± 0.21 | 12.22 ± 0.21 | 56.56 ± 0.93 | 14.85 ± 0.28 | 7.13 ± 0.14 | 11.61 ± 0.23 | 56.59 ± 0.71 | 12.07 ± 0.20 | 5.09 ± 0.14 | 10.33 ± 0.16 |
| CDFG | 49.83 ± 0.63 | 14.68 ± 0.32 | 6.50 ± 0.28 | 10.29 ± 0.22 | 54.21 ± 0.68 | 13.63 ± 0.40 | 5.02 ± 0.19 | 11.80 ± 0.38 | 56.18 ± 0.83 | 12.89 ± 0.19 | 5.37 ± 0.13 | 10.75 ± 0.13 | 50.25 ± 0.40 | 14.00 ± 0.34 | 6.35 ± 0.22 | 10.86 ± 0.22 |
| **DISH** | **65.72 ± 0.36** | **10.07 ± 0.16** | **4.01 ± 0.03** | **8.67 ± 0.26** | **64.50 ± 0.49** | 13.57 ± 0.01 | 6.37 ± 0.19 | **10.22 ± 0.25** | **66.43 ± 1.98** | **10.12 ± 0.97** | **3.71 ± 0.81** | **10.14 ± 0.45** | **68.33 ± 0.22** | **10.38 ± 0.22** | **3.89 ± 0.19** | **9.92 ± 0.22** |

## F.3 EVALUATION ON COMPLEX SENSITIVE STRUCTURES

Table 12: Performance comparison (%) with the state-of-the-art methods on UTKFace with code lengths varying from 16 to 128. Target Attribute: ethnicity, Sensitive Attribute: age(multi-class).

| Method | 16 bits | | | | 32 bits | | | | 64 bits | | | | 128 bits | | | |
|---|---|---|---|---|---|---|---|---|---|---|---|---|---|---|---|---|
| | MAP ↑ | EOD ↓ | EOP ↓ | DP ↓ | MAP ↑ | EOD ↓ | EOP ↓ | DP ↓ | MAP ↑ | EOD ↓ | EOP ↓ | DP ↓ | MAP ↑ | EOD ↓ | EOP ↓ | DP ↓ |
| OrthoHash | 60.39 ± 1.05 | 10.47 ± 0.68 | 6.04 ± 0.36 | 8.17 ± 0.72 | 62.76 ± 1.12 | 10.42 ± 0.74 | 6.20 ± 0.41 | 8.45 ± 0.80 | 64.93 ± 1.38 | 11.31 ± 0.89 | 6.44 ± 0.47 | 9.09 ± 0.86 | 64.98 ± 1.62 | 12.26 ± 0.97 | 6.98 ± 0.55 | 10.00 ± 0.93 |
| Bihalf | 54.10 ± 1.52 | 14.60 ± 2.05 | 9.80 ± 0.72 | 7.90 ± 1.08 | 56.20 ± 1.27 | 12.40 ± 1.84 | 8.10 ± 0.83 | 8.10 ± 1.02 | 57.10 ± 1.35 | 12.10 ± 1.76 | 7.60 ± 0.69 | 8.20 ± 1.15 | 56.80 ± 1.22 | 12.30 ± 1.88 | 7.90 ± 0.78 | 8.00 ± 1.07 |
| CE | 60.56 ± 1.42 | 13.38 ± 0.91 | 7.63 ± 0.49 | 9.63 ± 0.78 | 61.44 ± 1.19 | 11.72 ± 0.84 | 6.93 ± 0.45 | 8.70 ± 0.69 | 63.34 ± 1.27 | 13.71 ± 0.92 | 7.84 ± 0.52 | 10.41 ± 0.81 | 61.77 ± 1.05 | 13.12 ± 0.88 | 7.44 ± 0.47 | 9.85 ± 0.73 |
| CSQ | 65.51 ± 1.36 | 9.49 ± 0.78 | 5.58 ± 0.33 | 8.46 ± 0.62 | 64.61 ± 1.22 | 10.14 ± 0.71 | 5.96 ± 0.29 | 8.57 ± 0.59 | 64.31 ± 1.09 | 9.44 ± 0.67 | 5.42 ± 0.27 | 8.72 ± 0.55 | 63.86 ± 1.58 | 10.68 ± 0.83 | 6.33 ± 0.35 | 9.24 ± 0.71 |
| DFH | 50.93 ± 1.98 | 16.79 ± 1.21 | 8.47 ± 0.63 | 11.57 ± 0.97 | 65.02 ± 1.74 | 12.62 ± 1.08 | 7.35 ± 0.55 | 8.53 ± 0.88 | 59.08 ± 1.86 | 12.77 ± 0.96 | 7.32 ± 0.49 | 7.66 ± 0.79 | 67.84 ± 2.03 | 13.56 ± 1.15 | 7.98 ± 0.60 | 8.63 ± 0.92 |
| DPSH | 63.82 ± 1.14 | 10.61 ± 0.72 | 5.79 ± 0.32 | 9.61 ± 0.68 | 64.44 ± 1.06 | 11.34 ± 0.78 | 6.70 ± 0.36 | 9.43 ± 0.65 | 63.89 ± 1.21 | 11.85 ± 0.81 | 6.85 ± 0.40 | 10.01 ± 0.71 | 64.13 ± 1.33 | 12.60 ± 0.87 | 7.30 ± 0.43 | 10.36 ± 0.76 |
| DTSH | 63.66 ± 1.29 | 10.15 ± 0.66 | 5.34 ± 0.30 | 9.47 ± 0.61 | 65.37 ± 1.37 | 10.72 ± 0.73 | 6.28 ± 0.35 | 9.53 ± 0.67 | 62.64 ± 1.18 | 11.71 ± 0.79 | 7.14 ± 0.39 | 9.64 ± 0.69 | 61.30 ± 1.42 | 12.94 ± 0.85 | 7.94 ± 0.44 | 10.04 ± 0.74 |
| GreedyHash | 65.99 ± 1.35 | 11.00 ± 0.82 | 6.85 ± 0.41 | 8.98 ± 0.66 | 67.68 ± 1.48 | 11.36 ± 0.88 | 6.77 ± 0.38 | 9.58 ± 0.70 | 66.19 ± 1.27 | 12.12 ± 0.93 | 7.67 ± 0.43 | 9.48 ± 0.73 | 64.87 ± 1.52 | 11.26 ± 0.80 | 7.12 ± 0.40 | 9.09 ± 0.69 |
| SDH-C | 63.42 ± 1.41 | 11.90 ± 1.02 | 6.82 ± 0.37 | 9.41 ± 0.74 | 65.51 ± 1.19 | 12.68 ± 0.97 | 8.04 ± 0.45 | 9.48 ± 0.69 | 64.92 ± 1.23 | 14.02 ± 1.06 | 8.90 ± 0.49 | 9.65 ± 0.77 | 64.32 ± 1.36 | 15.99 ± 1.12 | 10.63 ± 0.53 | 10.18 ± 0.82 |
| DLBD | 30.20 ± 1.32 | 18.10 ± 1.42 | 10.70 ± 0.45 | 8.10 ± 0.56 | 30.40 ± 1.28 | 18.30 ± 1.39 | 10.90 ± 0.48 | 8.05 ± 0.52 | 31.80 ± 1.20 | 17.40 ± 1.27 | 10.20 ± 0.40 | 7.70 ± 0.49 | 32.30 ± 1.18 | 18.10 ± 1.86 | 10.60 ± 0.96 | 7.90 ± 0.93 |
| MDSHC | 62.20 ± 1.58 | 9.20 ± 1.10 | 6.20 ± 0.64 | 7.76 ± 0.37 | 64.80 ± 1.52 | 8.40 ± 0.72 | 5.50 ± 0.31 | 8.42 ± 0.59 | 63.90 ± 1.39 | 9.10 ± 0.61 | 6.80 ± 0.82 | 7.95 ± 0.31 | 63.10 ± 1.15 | 9.00 ± 1.02 | 5.90 ± 0.42 | 8.38 ± 0.25 |
| FATE | 65.26 ± 0.72 | 8.38 ± 0.72 | 4.65 ± 0.53 | 7.92 ± 0.34 | 67.72 ± 2.34 | 9.02 ± 0.84 | 4.75 ± 0.52 | 8.80 ± 0.63 | 67.62 ± 1.49 | 8.06 ± 0.64 | 3.99 ± 0.53 | 8.77 ± 0.33 | 67.66 ± 1.58 | 8.60 ± 0.83 | 4.15 ± 0.49 | 8.67 ± 0.47 |
| **DISH** | **70.54 ± 2.52** | **6.55 ± 0.66** | **2.81 ± 0.48** | **7.30 ± 0.39** | **70.57 ± 2.29** | **6.17 ± 0.94** | **2.70 ± 0.70** | **7.00 ± 0.36** | **71.12 ± 2.04** | **6.31 ± 0.49** | **2.96 ± 0.52** | **7.10 ± 0.32** | **71.75 ± 4.21** | **6.26 ± 0.40** | **2.58 ± 0.32** | **7.60 ± 0.46** |

Table 13: Performance comparison (%) with representative supervised hashing baselines and fair hashing methods on UTKFace with code lengths varying from 16 to 128. Target Attribute: age, Sensitive Attribute: ethnicity&gender.

| Method | 16 bits | | | | 32 bits | | | | 64 bits | | | | 128 bits | | | |
|---|---|---|---|---|---|---|---|---|---|---|---|---|---|---|---|---|
| | MAP ↑ | EOD ↓ | EOP ↓ | DP ↓ | MAP ↑ | EOD ↓ | EOP ↓ | DP ↓ | MAP ↑ | EOD ↓ | EOP ↓ | DP ↓ | MAP ↑ | EOD ↓ | EOP ↓ | DP ↓ |
| OrthoHash | 53.94 ± 1.21 | 16.01 ± 0.92 | 7.27 ± 0.41 | 11.05 ± 0.66 | 56.43 ± 1.18 | 15.88 ± 0.88 | 7.49 ± 0.39 | 11.15 ± 0.61 | 61.72 ± 1.35 | 14.54 ± 0.81 | 7.10 ± 0.37 | 10.64 ± 0.57 | 60.53 ± 1.29 | 17.05 ± 0.95 | 8.46 ± 0.44 | 11.85 ± 0.69 |
| Bihalf | 54.10 ± 1.82 | 21.35 ± 2.45 | 8.05 ± 0.52 | 15.20 ± 1.68 | 55.20 ± 1.76 | 20.40 ± 2.10 | 7.45 ± 0.60 | 14.70 ± 1.47 | 56.05 ± 1.63 | 19.80 ± 1.88 | 7.10 ± 0.49 | 14.20 ± 1.25 | 55.30 ± 1.57 | 18.90 ± 1.74 | 6.95 ± 0.51 | 13.30 ± 1.09 |
| CE | 59.09 ± 1.34 | 17.10 ± 0.89 | 8.81 ± 0.46 | 11.78 ± 0.71 | 60.06 ± 1.22 | 16.84 ± 0.83 | 7.44 ± 0.41 | 12.08 ± 0.68 | 59.85 ± 1.29 | 17.89 ± 0.91 | 8.27 ± 0.48 | 12.53 ± 0.75 | 60.49 ± 1.17 | 17.83 ± 0.86 | 8.40 ± 0.44 | 12.32 ± 0.70 |
| CSQ | 59.27 ± 1.26 | 16.96 ± 0.94 | 7.06 ± 0.39 | 12.89 ± 0.77 | 59.46 ± 1.19 | 16.85 ± 0.88 | 7.14 ± 0.36 | 12.74 ± 0.73 | 58.36 ± 1.33 | 16.26 ± 0.91 | 7.93 ± 0.43 | 10.78 ± 0.61 | 56.07 ± 1.41 | 16.51 ± 0.97 | 6.44 ± 0.34 | 13.06 ± 0.79 |
| DFH | 58.55 ± 1.52 | 13.36 ± 0.82 | 5.45 ± 0.36 | 10.91 ± 0.64 | 61.03 ± 1.47 | 15.42 ± 0.89 | 6.39 ± 0.40 | 12.04 ± 0.71 | 59.94 ± 1.39 | 14.91 ± 0.93 | 5.88 ± 0.38 | 12.10 ± 0.69 | 57.80 ± 1.62 | 15.60 ± 0.97 | 6.54 ± 0.42 | 11.93 ± 0.74 |
| DPSH | 57.59 ± 1.43 | 15.63 ± 0.86 | 6.72 ± 0.37 | 12.57 ± 0.69 | 57.25 ± 1.31 | 18.42 ± 0.98 | 8.83 ± 0.43 | 12.85 ± 0.78 | 58.30 ± 1.28 | 17.63 ± 0.92 | 8.18 ± 0.40 | 12.92 ± 0.73 | 59.68 ± 1.35 | 18.25 ± 1.01 | 8.41 ± 0.45 | 12.88 ± 0.80 |
| DTSH | 57.27 ± 1.29 | 15.57 ± 0.79 | 6.53 ± 0.33 | 12.52 ± 0.63 | 61.29 ± 1.44 | 15.32 ± 0.85 | 6.26 ± 0.36 | 12.66 ± 0.69 | 57.66 ± 1.21 | 15.98 ± 0.88 | 6.44 ± 0.35 | 12.03 ± 0.65 | 57.57 ± 1.33 | 15.22 ± 0.91 | 5.86 ± 0.32 | 12.60 ± 0.71 |
| GreedyHash | 65.03 ± 1.61 | 16.39 ± 0.92 | 7.44 ± 0.42 | 12.68 ± 0.72 | 60.39 ± 1.23 | 15.71 ± 0.87 | 7.13 ± 0.39 | 11.82 ± 0.67 | 63.58 ± 1.55 | 18.89 ± 1.02 | 9.26 ± 0.47 | 12.86 ± 0.79 | 60.33 ± 1.36 | 17.30 ± 0.96 | 7.58 ± 0.41 | 12.65 ± 0.75 |
| SDH-C | 62.72 ± 1.49 | 16.99 ± 1.08 | 7.95 ± 0.41 | 12.05 ± 0.69 | 62.26 ± 1.37 | 18.42 ± 1.15 | 8.53 ± 0.46 | 12.75 ± 0.76 | 62.41 ± 1.42 | 19.29 ± 1.21 | 9.51 ± 0.49 | 12.93 ± 0.81 | 61.93 ± 1.33 | 18.18 ± 1.09 | 8.80 ± 0.44 | 12.05 ± 0.73 |
| DLBD | 37.20 ± 1.65 | 22.10 ± 1.52 | 11.85 ± 1.18 | 11.25 ± 0.70 | 39.60 ± 1.57 | 22.45 ± 1.81 | 12.10 ± 1.05 | 11.60 ± 0.82 | 40.10 ± 1.49 | 23.30 ± 1.69 | 13.40 ± 0.92 | 10.50 ± 0.88 | 41.20 ± 1.41 | 24.05 ± 1.94 | 13.10 ± 0.95 | 11.90 ± 1.07 |
| MDSHC | 64.80 ± 1.32 | 14.20 ± 0.70 | 5.60 ± 0.40 | 9.25 ± 0.38 | 62.10 ± 1.25 | 13.40 ± 0.62 | 5.95 ± 0.37 | 9.48 ± 0.35 | 61.70 ± 1.18 | 12.30 ± 0.55 | 5.40 ± 0.39 | 10.95 ± 0.31 | 61.20 ± 1.10 | 12.55 ± 0.59 | 5.20 ± 0.35 | 10.60 ± 0.42 |
| FATE | 69.80 ± 0.65 | 8.67 ± 0.19 | 4.26 ± 0.19 | 8.38 ± 0.11 | 70.99 ± 0.67 | 8.81 ± 0.15 | 4.02 ± 0.13 | 8.98 ± 0.15 | 65.32 ± 0.45 | 9.91 ± 0.28 | 3.64 ± 0.23 | 10.08 ± 0.15 | 67.32 ± 0.99 | 11.11 ± 0.28 | 4.89 ± 0.13 | 9.71 ± 0.15 |
| **DISH** | **70.13 ± 1.91** | **7.41 ± 0.56** | **3.32 ± 0.26** | **7.91 ± 0.36** | **71.48 ± 1.94** | **7.87 ± 0.48** | **2.98 ± 0.29** | **8.86 ± 0.41** | **71.10 ± 1.21** | **8.77 ± 0.50** | **3.01 ± 0.53** | **9.52 ± 0.23** | **72.06 ± 2.18** | **7.67 ± 0.71** | **2.79 ± 0.39** | **8.60 ± 0.36** |

To demonstrate that DISH is not restricted to simple, binary sensitive attributes, we extend our evaluation on UTKFace to include more complex scenarios involving multi-class and intersectional

sensitive structures. These experiments test the framework's robustness when the sensitive groups are finer-grained or compositionally defined.

**Multi-class Sensitive Attribute (Age Groups).** We modify the standard setting (*Target: ethnicity, Sensitive: age*) by replacing the binary age split with a four-way classification based on quartiles (23, 29, and 45 years). This discretizes age into four distinct intervals, increasing the complexity of the debiasing task. As shown in Table 12, DISH consistently outperforms all baselines in retrieval accuracy (MAP) while achieving the lowest or near-lowest fairness gaps (EOD, EOP, DP) across all code lengths. This indicates that our channel-wise debiasing mechanism effectively generalizes to multi-class sensitive attributes without modification.

**Intersectional Sensitive Attribute (Ethnicity $\times$ Gender).** We further explore an intersectional setting (*Target: age, Sensitive: ethnicity & gender*) by defining the sensitive attribute as the Cartesian product of ethnicity and gender. This creates multiple intersectional subgroups (e.g., White-Male, Black-Female), introducing potential data imbalance and more challenging fairness constraints. The results in Table 13 demonstrate that DISH maintains the best fairness-accuracy trade-off, achieving higher MAP and consistently lower fairness disparities than both supervised baselines and the fair hashing method FATE. These findings confirm that DISH's factor-structured disentanglement is robust enough to handle multi-dimensional, intersectional bias.

