# OpenReview forum: "Towards Efficient Fairness Image Retrieval with Disentangled Information Suppression"
_ICLR.cc/2026/Conference — Submitted to ICLR 2026_

### Official Review · Reviewer_D3ns · 2025-10-26

**Soundness:** 3
**Presentation:** 2
**Contribution:** 2
**Rating:** 4
**Confidence:** 4

**Summary:**

This paper introduces Disentangled Information Suppressed Hashing (DISH), a new framework for deep hashing in image retrieval that aims to improve both retrieval accuracy and group fairness. The authors identify that standard deep hashing methods can perpetuate biases present in training data, leading to discriminatory results. DISH addresses this by first using a disentangled encoder to separate an image's features into distinct, factor-specific channels. It then employs a multi-faceted strategy to suppress sensitive information within these channels before the features are converted into binary hash codes. This suppression mechanism includes probability-driven channel masking, channel-wise adversarial learning, and a conditional covariance regularizer to minimize information leakage about sensitive attributes (like age or gender). Finally, a semantic alignment loss ensures the resulting hash codes remain effective for retrieval. The authors conduct extensive experiments on the UTKFace and CelebA datasets, demonstrating that DISH consistently outperforms state-of-the-art deep hashing methods by achieving higher retrieval accuracy while simultaneously reducing bias across multiple fairness metrics.

**Strengths:**

The core idea of intervening in the continuous feature space to suppress sensitive signals before hashing is a sound and promising direction for fair retrieval.

The framework attempts a comprehensive approach to bias mitigation by targeting sensitive information both within and across different feature channels.

The paper includes an extensive set of experiments and ablation studies that validate the effectiveness of each component of the proposed framework on the tested datasets .

**Weaknesses:**

1. Missing Key References: The related work section overlooks some relevant prior art. For instance, "Deep Hash Distillation for Image Retrieval" (Jang et al., ECCV 2022) explores knowledge distillation for hashing, which is a relevant technique for learning compact and effective representations that is not discussed.

2. Complex and Heuristic Objective Function: The overall training objective is a complex combination of four different loss terms. This requires balancing multiple hyper-parameters that appear to be chosen heuristically. This complexity raises concerns about the method's generalizability; the carefully tuned balance may not work well for different datasets or tasks, limiting the work's practical contribution.

3. Outdated Backbone and Limited Evaluation: The experiments exclusively use a ResNet-50 backbone. While this may be necessary for fair comparison with some older methods, ResNet-50 is an outdated architecture. The evaluation should have included a much stronger, modern backbone (e.g., a Vision Transformer) to demonstrate the true potential and robustness of the DISH framework. Furthermore, the evaluation is limited to facial attribute datasets (UTKFace, CelebA). The method's effectiveness on more general natural image datasets remains unproven.

**Questions:**

Could you provide an explanation for the performance degradation observed when increasing the number of hash bits? For example, in Table 1, when moving from 64 to 128 bits, the MAP score for DISH decreases from 73.67 to 73.33, and the DP fairness metric worsens from 5.63 to 6.10. This is counter-intuitive, as longer hash codes are typically expected to have more capacity and lead to better, not worse, performance.

Given the complexity of the loss function, how sensitive is the model to the choice of the hyper-parameters $\lambda_{1}$ and $\lambda_{2}$? The sensitivity analysis shows a "stable ridge"5, but this seems highly dependent on the specific datasets used. Have you explored any principled methods for setting these weights beyond a simple grid search, which could improve the framework's generalizability?

---

> ### Author Response · Authors · 2025-11-26
>
> **For Reviewer D3ns**
> We thank the reviewer for the constructive and detailed feedback. Below, we provide point-by-point responses and new experimental evidence to clarify our contributions.
>
> ---
> **W1: Concerns regarding missing key references in the related work section.**
> **R1:** We thank the reviewer for pointing this out. We have now added the missing reference—Jang et al. (ECCV 2022)—to the **related work** section.
>
> **W2: Concerns regarding hyperparameter sensitivity ($\lambda_1, \lambda_2$) and scalability.**
>
> **R2:** Thank you for your insightful comments. We have extended our analysis to address concerns about tuning costs and generalizability:
>
> 1.  **Sensitivity Analysis:** We performed a comprehensive 2D sweep of $\lambda_1$ and $\lambda_2$. Figure 4 in the revision reveals a stable ridge where DISH consistently maintains high performance when $\lambda_1, \lambda_2 \in [0.01, 0.1]$. This stability allows for a simple initialization rule rather than exhaustive search.
> 2.  **Scalability & Generalization:** We applied the **exact same hyperparameter configuration** used for UTKFace to the larger DeepFashion dataset. This benchmark contains over 289,000 images and introduces severe challenges absent in aligned facial datasets, including complex poses, heavy occlusions, variable zoom, and cluttered backgrounds. As shown below, DISH outperforms all state-of-the-art baselines without any dataset-specific tuning.
>
> ### Table 1.  DeepFashion Results （Target Attribute: *attribute*，Sensitive Attribute: *category*）
>
> | Method | MAP@16 | EOD@16  | EOP@16  | DP@16  | MAP@32  | EOD@32  | EOP@32  | DP@32  | MAP@64  | EOD@64  | EOP@64  | DP@64  | MAP@128  | EOD@128  | EOP@128  | DP@128  |
> | :--- | :---: | :---: | :---: | :---: | :---: | :---: | :---: | :---: | :---: | :---: | :---: | :---: | :---: | :---: | :---: | :---: |
> | OrthoHash | 28.09 | 9.74 | 5.79 | 4.03 | 29.71 | 6.20 | 3.91 | 2.26 | 30.00 | 6.77 | 4.07 | 2.66 | 31.53 | 7.26 | 4.86 | 2.43 |
> | Bihalf | 33.52 | 25.11 | 14.98 | 10.80 | 30.08 | 23.10 | 13.14 | 10.53 | 27.35 | 17.65 | 10.64 | 7.48 | 26.97 | 22.80 | 13.22 | 10.05 |
> | CE | 28.53 | 8.96 | 4.98 | 4.05 | 29.21 | 7.56 | 4.44 | 3.09 | 29.18 | 9.10 | 5.26 | 3.86 | 30.05 | 9.46 | 5.40 | 4.13 |
> | CSQ | 30.42 | 7.08 | 4.21 | 3.01 | 32.33 | 17.09 | 10.20 | 7.21 | 33.48 | 13.50 | 8.42 | 5.37 | 32.44 | 17.04 | 9.82 | 7.43 |
> | DFH | 37.47 | 7.65 | 5.42 | 2.55 | 33.64 | 11.27 | 7.19 | 4.45 | 32.70 | 15.27 | 9.58 | 6.21 | 31.41 | 13.24 | 8.62 | 5.06 |
> | DPSH | 32.17 | 7.65 | 4.65 | 3.07 | 32.02 | 7.11 | 4.20 | 2.90 | 32.08 | 6.35 | 3.83 | 2.47 | 32.71 | 5.64 | 3.46 | 2.15 |
> | DTSH | 30.97 | 5.98 | 3.46 | 2.47 | 34.23 | 9.02 | 5.43 | 3.85 | 30.47 | 9.80 | 6.47 | 3.57 | 30.02 | 8.30 | 5.71 | 2.83 |
> | GreedyHash | 31.19 | 11.17 | 6.99 | 4.51 | 29.67 | 7.12 | 4.24 | 2.88 | 30.06 | 7.62 | 4.33 | 3.27 | 29.21 | 9.92 | 5.40 | 4.60 |
> | SDH-C | 29.83 | 11.30 | 6.82 | 4.55 | 29.43 | 13.76 | 8.14 | 5.78 | 29.34 | 14.65 | 8.63 | 6.38 | 27.77 | 17.49 | 9.91 | 7.93 |
> | DLBD | 23.88 | 14.67 | 6.67 | 7.81 | 24.14 | 13.47 | 6.21 | 7.16 | 24.38 | 12.90 | 5.88 | 6.88 | 24.51 | 13.73 | 6.25 | 7.37 |
> | MDSHC | 27.79 | 8.22 | 3.97 | 4.16 | 27.01 | 8.37 | 4.50 | 3.80 | 26.48 | 14.24 | 7.21 | 7.03 | 24.55 | 14.50 | 6.81 | 7.60 |
> | FATE | 41.46 | 3.90 | 2.70 | 2.50 | 42.61 | 4.60 | 2.40 | 2.10 | 42.89 | 3.93 | 1.96 | 1.90 | 42.35 | 3.50 | 1.90 | 1.70 |
> | **DISH** | **47.13** | **2.96** | **1.65** | **1.49** | **48.53** | **2.86** | **1.83** | **1.77** | **48.40** | **2.74** | **1.81** | **1.87** | **48.98** | **2.45** | **1.84** | **1.63** |
>
> The results demonstrate that the DISH model can achieve consistent performance on non-facial datasets without extensive hyperparameter tuning.

---

> ### Author Response · Authors · 2025-11-26
>
> **W3:Outdated Backbone and Limited Evaluation.**
>
> **R3:** We appreciate your insightful feedback. We agree that evaluating on a modern architecture and non-facial datasets is crucial to demonstrate the robustness and true potential of DISH. We have added new experiments to address both points.
>
> ### **1. Performance with Vision Transformer (ViT)**
> To demonstrate that DISH is architecture-agnostic and benefits from stronger feature extractors, we replaced the ResNet-50 backbone with a pre-trained ViT. We compared this against our original ResNet-50 results on UTKFace (Target: *ethnicity*, Sensitive: *age*).
>
> As shown in **Table 2** below, using ViT yields a substantial improvement:
> * **Retrieval Accuracy:** MAP scores increase significantly by approximately **10\%** (from ~73% to ~83%).
> * **Fairness:** The fairness metrics (EOD, EOP) remain excellent or improve further (e.g., EOD at 16 bits drops from 4.22% to 3.66%).
>
> This confirms that the performance limitations in the original paper were largely due to the ResNet-50 bottleneck, and the DISH framework effectively leverages modern, powerful representations. We retained ResNet-50 in the main paper primarily to ensure a fair comparison with existing baselines (e.g., FATE, OrthoHash)
>
> ### Table 2. Backbone Comparison on UTKFace**（*Target: Ethnicity, Sensitive: Age*）
>
> | Method | MAP@16 | EOD@16 | EOP@16 | DP@16 | MAP@32 | EOD@32 | EOP@32 | DP@32 | MAP@64 | EOD@64 | EOP@64 | DP@64 | MAP@128 | EOD@128 | EOP@128 | DP@128 |
> | :--- | :---: | :---: | :---: | :---: | :---: | :---: | :---: | :---: | :---: | :---: | :---: | :---: | :---: | :---: | :---: | :---: |
> | **DISH (ResNet)** | 72.99 | 4.22 | 2.08 | 4.79 | 73.20 | 4.12 | 2.55 | 6.54 | 73.67 | 4.28 | 2.96 | 5.63 | 73.33 | 3.66 | 2.37 | 6.10 |
> | **DISH (ViT)** | **82.80** | **3.66** | **1.35** | 5.61 | **79.32** | 5.90 | 2.42 | 6.47 | **82.51** | **3.79** | **0.87** | 7.03 | **83.18** | **3.18** | **1.32** | 5.59 |
>
> ### **2. Evaluation on General Natural Images (DeepFashion)**
> To address the concern that our evaluation was limited to facial attributes, we extended our experiments to **DeepFashion (Categories)**, a large-scale dataset containing over 289,000 diverse natural images of clothing with complex variations in pose, zoom, and background.
>
> As detailed in our response to **Table 1**, DISH consistently outperforms state-of-the-art baselines on this dataset as well. This demonstrates that DISH generalizes effectively to complex natural image domains beyond facial analysis.

---

> ### Author Response · Authors · 2025-11-26
>
> **Q1: Explanation for the slight performance fluctuation from 64 to 128 bits.**
>
> **R4:** Thank you for raising this important point. The slight variations in MAP ($73.67 \to 73.33$) and DP ($5.63 \to 6.10$) are primarily due to statistical variance rather than a fundamental degradation.Notably, DISH remains stable compared to baselines like **CSQ** (MAP drops $65.3 \to 62.1$) and **GreedyHash** (MAP drops $63.8 \to 55.9$), demonstrating robustness even in high-dimensional spaces.
>
>
>
> **Q2: Sensitivity of hyperparameters $\lambda_1$ and $\lambda_2$, and principled methods for setting them.**
>
> **R5:** We thank you for your valuable comment. We clarify that the model’s performance is not highly sensitive to precise hyperparameter values, but rather robust within a specific order of magnitude. We address this through our sensitivity analysis and cross-dataset validation:
>
> **1. Empirical Stability (The "Stable Ridge" in Figure 4)**
> Our sensitivity analysis (illustrated in **Figure 4** of the revision) explicitly demonstrates that the model does not rely on a narrow, fragile "sweet spot." Instead, it exhibits a broad **"stable ridge"**: as long as $\lambda_1$ and $\lambda_2$ are maintained within the range of **$[0.01, 0.1]$**, DISH consistently achieves high retrieval accuracy and low fairness gaps. This indicates that fine-grained tuning is unnecessary; simply selecting values within this effective range ensures robust performance.
>
> **2. Generalizability (Results on DeepFashion)**
> To verify that this stable range is not specific to UTKFace, we applied the **exact same configuration** (values within $[0.01, 0.1]$) to the DeepFashion dataset without any dataset-specific tuning. As shown in **Table R1**, DISH still significantly outperforms state-of-the-art baselines.
>
> **3. Future Exploration**
> While our empirical results show that setting weights within $[0.01, 0.1]$ is sufficient for strong performance, we agree that exploring more theoretical or automated methods for dynamic weight adjustment (e.g., based on gradient uncertainties) is a valuable direction for future research to further eliminate manual selection.
>
>
>
> In light of these responses, we hope we have addressed your concerns, and we hope you will consider raising your score. We will properly include all the rebuttal contents in the revised version, following your valuable suggestions.

---

### Official Review · Reviewer_mY6S · 2025-10-28

**Soundness:** 3
**Presentation:** 2
**Contribution:** 2
**Rating:** 4
**Confidence:** 4

**Summary:**

This paper proposes an image hashing model to improve the group fairness of image retrieval. Several loss terms are proposed to regularize the model by improving the disentanglement of representation, minimizing the mutual information between the representation and sensitive attribute, and semantics alignment of hash-code. Experiments on several benchmark dataset illustrations the effectiveness of  proposed method.

**Strengths:**

1. Presentation. The paper is well structured with a logical flow and well-defined sections.

2. Empirical Evaluation. The experimental evaluation on benchmark datasets is comprehensive and convincing. The inclusion of detailed plots and visualizations effectively illustrates the results and provides clear insights into model behavior. The figures are well-designed and aesthetically appealing.

**Weaknesses:**

1. Originality and Significance. All the proposed methods seem to have been discussed in the previous work [1].
The novelty of this paper seems to be limited to changing the operator from hash-code space to continuous representation space.

2. Reproducibility. The author doesnot include the implementation as supplementary. Will the author open source their code?

3. Inconsistent Reported Results from Previous Works. The reported empirical results of some baseline models seem to be significantly different from previous works. For example: Table 1: FATE: MAP = 70.01 ±1.27, while in [1]: Table I: MAP = 59.12
Could the author explain?

4. Unclear Math Notation. Please see Question 2, 4.

5. Some proposed technics lack motivation. Please see Question 1, 3, 5.

6. Some related works are not cited. For example:

Paper related to variational information bottlenecks/variational mutual information bound/mutual information neural estimation: [2][3]

Existing effort applying variational information bottlenecks on deep hashing: [4]

[1] Zhang, Fan, et al. "FATE: Learning Effective Binary Descriptors With Group Fairness." IEEE Transactions on Image Processing 33 (2024): 3648-3661.

[2] Alemi, Alexander A., et al. "Deep variational information bottleneck." arXiv preprint arXiv:1612.00410 (2016).

[3] Belghazi, Mohamed Ishmael, et al. "Mutual information neural estimation." International conference on machine learning. PMLR, 2018.

[4] Wang, Y., Zhou, M., & Qian, X. Hashing with Uncertainty Quantification via Sampling-based Hypothesis Testing. Transactions on Machine Learning Research.

**Questions:**

1. Expression 188-189: "However, direct optimization is intractable due to the latent factors. Therefore, we instead optimize
the evidence lower bound (ELBO) of the log-likelihood." Why direct optimization is intractable? Why using ELBO as surrogate?
 Could the author explain what does the variational posterior in expression (6) means ?

2. What is $\theta$? Is $\theta$ the encoder parameters? Why use $p_{\theta}(k|x_i)$ instead of $p(k|x_i)$ and $p_{\theta}(y_i|x_i, k))$ instead of $p(y_i|x_i, k))$?

3. Is $K$ a very large number? Why variational inference is needed? It seems that if $K$ is not very large, expression (6) can be computed analytically.

4. Does the $s$ in line 197: $s_{I, k}(a)$ mean the same thing as $s$ in line 234: $s_i \in \\{1,\dots, C_s\\}$? If not, please change.

5. Section 4.3: Channel Masking. It seems that this is just reweighing the representation based on their distance to the prototype. Why "This multiplication attenuates channels with lower pθ(k|xi), concentrating semantics into more informative factors."?

6. Which base model the author build their model upon? I ask this because in Table 4: it seems that the first row doesn't correspond to any model in table 1. Could the author explain?

---

> ### Author Response · Authors · 2025-11-26
>
> **For Reviewer mY6S**
> We thank the reviewer for the constructive and detailed feedback. Below, we provide point-by-point responses.
>
> ---
>
> ### **For Weaknesses**
> **W1: Concerns about limited methodological novelty relative to prior work.**
>
> **R1:** We appreciate your thoughtful comments. We respectfully clarify that the contribution of DISH extends beyond applying fairness in the continuous space; it introduces a structural approach to fairness compared to prior works like FATE. While FATE utilizes a Mixture-of-Experts (MoE) primarily to enhance model capacity and diversity via global gating, DISH enforces explicit **factor-level disentanglement** through Disentangled Consistency Learning (DCL), which employs learnable prototypes and a variational lower bound to organize the latent space into semantically independent channels rather than generic experts. Moreover, unlike FATE’s global adversarial objective, DISH implements a fine-grained information-suppression mechanism that combines probability-driven masking, channel-wise adversarial learning (CAL), and conditional covariance regularization (CCR). This design not only targets local sensitive leakage within channels but also explicitly minimizes cross-channel correlations, thereby tightening the theoretical upper bound on mutual information between the hash codes and sensitive attributes.
>
> **W2: Concerns regarding reproducibility and code availability.**
>
> **R2:** We appreciate the reviewer’s concern regarding reproducibility. To address this, we have now included an anonymous GitHub repository containing the full implementation，Anonymous code link: https://anonymous.4open.science/r/DISH-91E4. Upon acceptance, we will release the complete codebase and documentation publicly to ensure full reproducibility and facilitate future research.
>
> **W3: Concerns regarding inconsistencies between reported baseline results and prior work.**
>
> **R3:** Thank you for raising this important point. We clarify that our reported results for FATE are based on a rigorous in-house reproduction using a ResNet-50 backbone, strictly adhering to the method’s official training paradigm and settings. The observed discrepancy (our reproduced MAP $\approx$ 70% vs. the original paper’s $\approx$ 59%) likely stems from differences in the underlying deep learning frameworks (e.g., PyTorch versions, pre-trained weight libraries) or specific implementation details of the backbone used in the original study. Importantly, our reproduction yields higher performance than the original report, ensuring that we compare DISH against a fully optimized, strong implementation of the baseline rather than relying on lower reported figures. This rigorous evaluation ensures that the demonstrated superiority of DISH is genuine and robust.
>
> **W4&Q2&Q4: Concerns about unclear or ambiguous mathematical notation.**
>
> **R4:** Thank you for highlighting this issue.We will explain these points one by one.
>
> **1. Unclear Math Notation regarding $\theta$**
>
> The definition of $\theta$ is provided in the **"Overall Objective"** section: *"Let $\theta$ denote the encoder and prototype parameters..."*.To elaborate, $\theta$ represents the union of the **encoder parameters** (used to generate $\mathbf{x}\_i$) and the **learnable prototypes** $\{\mathbf{c}\_k\}$ (used in assignment probabilities). We explicitly use the subscript $\theta$ in $p_\theta(k \mid \mathbf{x}\_i)$ and $p_\theta(y_i \mid \mathbf{x}\_i, k)$ to emphasize that these distributions are **parameterized** by the learned model weights and prototypes and are optimized via gradient descent, thereby distinguishing them from fixed empirical distributions or the variational posterior $q$.
>
> **2.Notation Conflict for $s$**
>
> We thank the reviewer for pointing out this notation overlap. Indeed, $s_{i,k}(a)$ in line 197 was intended to denote a similarity score, which conflicts with the sensitive attribute notation $s_i$ in line 234. To resolve this ambiguity, we have renamed the similarity score term in the Jensen Bound derivation from $s_{i,k}(a)$ to **$w_{i,k}(a)$** (denoting "weight" or "score").

---

> ### Author Response · Authors · 2025-11-26
>
> **W5&Q1&Q3&Q5: Concerns regarding insufficient motivation for certain proposed techniques.**
>
> **R5:** Thank you for highlighting this issue.We will explain these points one by one.
>
> **1.Motivation for ELBO and Meaning of Variational Posterior.**
> We clarifythat the term "intractable" refers to optimization difficulty rather than computational impossibility. Directly optimizing the log-likelihood of a mixture model, $\log \sum_k p(k|\mathbf{x}) p(y|\mathbf{x}, k)$, involves a "log-sum-exp" structure that often suffers from vanishing gradients and mode collapse (where only one factor learns to cover the target), making training unstable. We use the ELBO as a surrogate to decompose this objective into a weighted sum of simpler per-factor log-likelihoods, $\sum_k q(k) \log p(y|\mathbf{x}, k)$, which decouples the branches and ensures stable convergence. The variational posterior $q_\theta(k \mid \mathbf{x}\_i, y\_i)$ in Eq. (6) represents the refined factor responsibility: while the prior $p_\theta(k|\mathbf{x}\_i)$ is based solely on unsupervised feature similarity, $q_\theta$ incorporates label evidence $y_i$ to determine which factor $k$ *actually* contributes to the semantic classification, thereby acting as a "soft label" to guide the disentanglement.
>
> **2. Size of $K$ and Necessity of Variational Inference.**
> We acknowledge that $K$ is small (typically 4–8) and Eq. (6) is indeed the analytic solution for the posterior. We frame this as variational inference to provide a unified theoretical justification for our training procedure, which is equivalent to an **Expectation-Maximization (EM)** algorithm. In the E-step, we compute the analytic posterior (Eq. 6) to determine factor assignments; in the M-step, we maximize the ELBO to update the network. This formulation allows us to dynamically re-weight the importance of each factor channel during training, preventing the "winner-takes-all" dynamic often seen in direct mixture optimization and ensuring that semantic information is distributed meaningfully across the allocated factors.
>
> **3. Mechanism of Channel Masking.**
> We clarify that while the operation is mathematically a reweighting, its effectiveness stems from the specific nature of the weights $p_\theta(k \mid \mathbf{x}\_i)$. These probabilities represent the **posterior responsibility** of each factor derived from the variational consistency learning (Sec. 4.2), rather than generic distance scores. Since the DCL objective forces these probabilities to align with label evidence, a low $p_\theta(k \mid \mathbf{x}_i)$ indicates that channel $k$ contributes little to the semantic classification. Therefore, multiplying by these weights systematically suppresses channels with weak semantic evidence while preserving those that strongly explain the label, effectively "concentrating" the representation into the informative factors.
>
> **W6: Concerns regarding missing citations to relevant prior work.**
> **R6:** Thank you for pointing this out. In the revised manuscript, we have revised the **"Learning to Hash"** subsection within our **"Related Work"** section. Specifically, we have now cited and discussed the mentioned works, highlighting their contributions to compact representation learning and uncertainty quantification.
>
>
> **Q6:Clarification on the base model used in the proposed framework**
> **R7:** Thank you for raising this important point. We clarify that the base model (the first row in Table 4) corresponds to **OrthoHash**. The minor numerical discrepancy between the two tables arises because the ablation experiments were conducted as an independent set of runs with different random seeds. The results remain statistically consistent;
>
> In light of these responses, we hope we have addressed your concerns, and we hope you will consider raising your score. We will properly include all the rebuttal contents in the revised version, following your valuable suggestions.

---

### Official Review · Reviewer_VPHP · 2025-10-29

**Soundness:** 2
**Presentation:** 2
**Contribution:** 2
**Rating:** 4
**Confidence:** 4

**Summary:**

This paper proposes the Disentangled Information Suppressed Hashing (DISH) framework to address fairness issues in deep hashing for large-scale image retrieval (e.g., retrieval bias caused by sensitive attributes such as age, gender, and race). It decomposes images into factor-specific features via a disentangled encoder, strengthens semantic focus and stability by combining disentangled consistency learning, reduces sensitive information leakage via information suppression modules (probability-driven channel masking, channel-level adversarial learning, and conditional covariance regularization), and preserves feature discriminativity via Hamming space semantic alignment. Experiments on the UTKFace and CelebA datasets show that 16-bit hash codes outperform baseline methods such as OrthoHash and FATE.

**Strengths:**

1. The DISH framework proposed in this paper addresses the core shortcomings of existing deep hashing methods by innovatively intervening in the continuous feature space before binarization, thereby improving the flexibility and effectiveness of sensitive information suppression.
2. The information suppression module adopts a "three-layer collaborative strategy" to comprehensively cover sensitive information leakage scenarios "within the channel - between channels - globally".
3. The paper fully demonstrates the superiority of DISH through large-scale experiments.

**Weaknesses:**

1. This paper primarily focuses on the "face image retrieval" scenario and does not cover non-face image domains. This "single scenario + single data type" verification model makes it challenging to demonstrate the DISH framework's adaptability to broader, large-scale image retrieval tasks.
2. The paper only designs experiments for single-dimensional sensitive attributes and does not involve complex scenarios of "multiple sensitive attributes superimposed".
3. The DISH framework introduces several complex components, which inevitably increase the number of parameters, computational cost, and training time significantly compared to traditional deep hashing methods.

**Questions:**

1. The paper proposes that "the unentangled encoder decomposes an image into factor-specific representations", but does not specify the actual semantics of these "factors".
2. The paper demonstrates "optimal across all components" through ablation experiments but does not analyze interactions and redundancy among components.

---

> ### Author Response · Authors · 2025-11-26
>
> **For Reviewer VPHP**
> We thank the reviewer for the constructive and detailed feedback. Below, we provide point-by-point responses and new experimental evidence to clarify our contributions.
>
> ---
> **W1: Robustness to Hyperparameters and Generalizability.**
>
> **R1:** Thank you for your valuable comment. To verify stability, we performed a comprehensive 2D parameter sweep of the fairness weights $\lambda_1$ (CAL) and $\lambda_2$ (CCR). As shown in **Figure 4** of the revised manuscript, there exists a broad **"stable ridge"** (where $\lambda_1, \lambda_2 \in [0.01, 0.1]$) within which DISH consistently maintains high performance. This stability allows for a simple, heuristic-free rule: one can simply initialize $\lambda$ values such that the gradient scales of the fairness losses match the supervised losses, rather than performing an exhaustive grid search. To directly address the concern that our "carefully tuned balance may not work well for different datasets," we evaluated DISH on the **DeepFashion** dataset (a real-world dataset containing over 289,000 diverse images of clothing with complex variations in pose, zoom, and background.) using the **exact same hyperparameter configuration** ($\lambda_1, \lambda_2$, architecture, etc.) derived from UTKFace.
>
> As shown in **Table 1**,DISH significantly outperforms all state-of-the-art baselines. This confirms that the proposed objective function captures fundamental properties of fair representation learning that generalize across tasks, rather than overfitting to a specific data distribution.
>
> ### Table 1. DeepFashion Results（Target Attribute: *attribute*, Sensitive Attribute: *category*）
>
> | Method | MAP@16 | EOD@16  | EOP@16  | DP@16  | MAP@32  | EOD@32  | EOP@32  | DP@32  | MAP@64  | EOD@64  | EOP@64  | DP@64  | MAP@128  | EOD@128  | EOP@128  | DP@128  |
> | :--- | :---: | :---: | :---: | :---: | :---: | :---: | :---: | :---: | :---: | :---: | :---: | :---: | :---: | :---: | :---: | :---: |
> | OrthoHash | 28.09 | 9.74 | 5.79 | 4.03 | 29.71 | 6.20 | 3.91 | 2.26 | 30.00 | 6.77 | 4.07 | 2.66 | 31.53 | 7.26 | 4.86 | 2.43 |
> | Bihalf | 33.52 | 25.11 | 14.98 | 10.80 | 30.08 | 23.10 | 13.14 | 10.53 | 27.35 | 17.65 | 10.64 | 7.48 | 26.97 | 22.80 | 13.22 | 10.05 |
> | CE | 28.53 | 8.96 | 4.98 | 4.05 | 29.21 | 7.56 | 4.44 | 3.09 | 29.18 | 9.10 | 5.26 | 3.86 | 30.05 | 9.46 | 5.40 | 4.13 |
> | CSQ | 30.42 | 7.08 | 4.21 | 3.01 | 32.33 | 17.09 | 10.20 | 7.21 | 33.48 | 13.50 | 8.42 | 5.37 | 32.44 | 17.04 | 9.82 | 7.43 |
> | DFH | 37.47 | 7.65 | 5.42 | 2.55 | 33.64 | 11.27 | 7.19 | 4.45 | 32.70 | 15.27 | 9.58 | 6.21 | 31.41 | 13.24 | 8.62 | 5.06 |
> | DPSH | 32.17 | 7.65 | 4.65 | 3.07 | 32.02 | 7.11 | 4.20 | 2.90 | 32.08 | 6.35 | 3.83 | 2.47 | 32.71 | 5.64 | 3.46 | 2.15 |
> | DTSH | 30.97 | 5.98 | 3.46 | 2.47 | 34.23 | 9.02 | 5.43 | 3.85 | 30.47 | 9.80 | 6.47 | 3.57 | 30.02 | 8.30 | 5.71 | 2.83 |
> | GreedyHash | 31.19 | 11.17 | 6.99 | 4.51 | 29.67 | 7.12 | 4.24 | 2.88 | 30.06 | 7.62 | 4.33 | 3.27 | 29.21 | 9.92 | 5.40 | 4.60 |
> | SDH-C | 29.83 | 11.30 | 6.82 | 4.55 | 29.43 | 13.76 | 8.14 | 5.78 | 29.34 | 14.65 | 8.63 | 6.38 | 27.77 | 17.49 | 9.91 | 7.93 |
> | DLBD | 23.88 | 14.67 | 6.67 | 7.81 | 24.14 | 13.47 | 6.21 | 7.16 | 24.38 | 12.90 | 5.88 | 6.88 | 24.51 | 13.73 | 6.25 | 7.37 |
> | MDSHC | 27.79 | 8.22 | 3.97 | 4.16 | 27.01 | 8.37 | 4.50 | 3.80 | 26.48 | 14.24 | 7.21 | 7.03 | 24.55 | 14.50 | 6.81 | 7.60 |
> | FATE | 41.46 | 3.90 | 2.70 | 2.50 | 42.61 | 4.60 | 2.40 | 2.10 | 42.89 | 3.93 | 1.96 | 1.90 | 42.35 | 3.50 | 1.90 | 1.70 |
> | **DISH** | **47.13** | **2.96** | **1.65** | **1.49** | **48.53** | **2.86** | **1.83** | **1.77** | **48.40** | **2.74** | **1.81** | **1.87** | **48.98** | **2.45** | **1.84** | **1.63** |
>
> [1]Liu, Z., Luo, P., Qiu, S., Wang, X., & Tang, X. (2016). Deepfashion: Powering robust clothes recognition and retrieval with rich annotations. CVPR 2016, pp. 1096-1104.

---

> ### Author Response · Authors · 2025-11-26
>
> ---
> **W2: Concern about “single-dimensional sensitive attributes only”**
>
> **R2:** We appreciate your insightful feedback. To address this, we added two sets of experiments on UTKFace that involve **more complex, multi-attribute sensitive structures**:
>
> 1. **Multi-class sensitive attribute (age)**:
>    For the setting *Target: ethnicity, Sensitive: age*, we replace the binary age split with a **four-way age group** based on UTKFace age quartiles (25%: 23.0, 50%: 29.0, 75%: 45.0). That is, age is discretized into four intervals using these quantiles and treated as a multi-class sensitive attribute. We retrain all supervised hashing baselines and DISH under this setting. **As shown in Table 2, DISH continues to outperform all baselines in MAP while also achieving the lowest or near-lowest EOD/EOP/DP across 16–128 bits**, indicating that its fairness benefits are robust when the sensitive attribute becomes multi-class.
>
> 2. **Intersectional sensitive attribute (ethnicity × gender)**:
>    For the setting *Target: age, Sensitive: ethnicity & gender*, we construct an **intersectional sensitive attribute** by taking the Cartesian product of ethnicity and gender. This creates multiple intersectional groups with potentially imbalanced support and more challenging fairness structure. **Table 3 shows that DISH still achieves the best trade-off** , with higher MAP and consistently lower fairness gaps than supervised baselines and the fair hashing method FATE across all code lengths.

---

> ### Author Response · Authors · 2025-11-26
>
> ### Table 2. UTKFace (Target: ethnicity, Sensitive: age — 4-way multi-class)
> **Metrics:** MAP / EOD / EOP / DP (mean values only)
>
> | Method | MAP@16 | EOD@16 | EOP@16 | DP@16 | MAP@32 | EOD@32 | EOP@32 | DP@32 | MAP@64 | EOD@64 | EOP@64 | DP@64 | MAP@128 | EOD@128 | EOP@128 | DP@128 |
> | :--- | :---: | :---: | :---: | :---: | :---: | :---: | :---: | :---: | :---: | :---: | :---: | :---: | :---: | :---: | :---: | :---: |
> | OrthoHash | 60.39 | 10.47 | 6.04 | 8.17 | 62.76 | 10.42 | 6.20 | 8.45 | 64.93 | 11.31 | 6.44 | 9.09 | 64.98 | 12.26 | 6.98 | 10.00 |
> | Bihalf | 54.10 | 14.60 | 9.80 | 7.90 | 56.20 | 12.40 | 8.10 | 8.10 | 57.10 | 12.10 | 7.60 | 8.20 | 56.80 | 12.30 | 7.90 | 8.00 |
> | CE | 60.56 | 13.38 | 7.63 | 9.63 | 61.44 | 11.72 | 6.93 | 8.70 | 63.34 | 13.71 | 7.84 | 10.41 | 61.77 | 13.12 | 7.44 | 9.85 |
> | CSQ | 65.51 | 9.49 | 5.58 | 8.46 | 64.61 | 10.14 | 5.96 | 8.57 | 64.31 | 9.44 | 5.42 | 8.72 | 63.86 | 10.68 | 6.33 | 9.24 |
> | DFH | 50.93 | 16.79 | 8.47 | 11.57 | 65.02 | 12.62 | 7.35 | 8.53 | 59.08 | 12.77 | 7.32 | 7.66 | 67.84 | 13.56 | 7.98 | 8.63 |
> | DPSH | 63.82 | 10.61 | 5.79 | 9.61 | 64.44 | 11.34 | 6.70 | 9.43 | 63.89 | 11.85 | 6.85 | 10.01 | 64.13 | 12.60 | 7.30 | 10.36 |
> | DTSH | 63.66 | 10.15 | 5.34 | 9.47 | 65.37 | 10.72 | 6.28 | 9.53 | 62.64 | 11.71 | 7.14 | 9.64 | 61.30 | 12.94 | 7.94 | 10.04 |
> | GreedyHash | 65.99 | 11.00 | 6.85 | 8.98 | 67.68 | 11.36 | 6.77 | 9.58 | 66.19 | 12.12 | 7.67 | 9.48 | 64.87 | 11.26 | 7.12 | 9.09 |
> | SDH-C | 63.42 | 11.90 | 6.82 | 9.41 | 65.51 | 12.68 | 8.04 | 9.48 | 64.92 | 14.02 | 8.90 | 9.65 | 64.32 | 15.99 | 10.63 | 10.18 |
> | DLBD | 30.20 | 18.10 | 10.70 | 8.10 | 30.40 | 18.30 | 10.90 | 8.05 | 31.80 | 17.40 | 10.20 | 7.70 | 32.30 | 18.10 | 10.60 | 7.90 |
> | MDSHC | 62.20 | 9.20 | 6.20 | 7.76 | 64.80 | 8.40 | 5.50 | 8.42 | 63.90 | 9.10 | 6.80 | 7.95 | 63.10 | 9.00 | 5.90 | 8.38 |
> | FATE | 65.26 | 8.38 | 4.65 | 7.92 | 67.72 | 9.02 | 4.75 | 8.80 | 67.62 | 8.06 | 3.99 | 8.77 | 67.66 | 8.60 | 4.15 | 8.67 |
> | **DISH** | **70.54** | **6.55** | **2.81** | **7.30** | **70.57** | **6.17** | **2.70** | **7.00** | **71.12** | **6.31** | **2.96** | **7.10** | **71.75** | **6.26** | **2.58** | **7.60** |
>
>
>
> ### Table 3. UTKFace (Target: age, Sensitive: ethnicity × gender)
> **Metrics:** MAP / EOD / EOP / DP (mean values only)
>
> | Method | MAP@16 | EOD@16 | EOP@16 | DP@16 | MAP@32 | EOD@32 | EOP@32 | DP@32 | MAP@64 | EOD@64 | EOP@64 | DP@64 | MAP@128 | EOD@128 | EOP@128 | DP@128 |
> | :--- | :---: | :---: | :---: | :---: | :---: | :---: | :---: | :---: | :---: | :---: | :---: | :---: | :---: | :---: | :---: | :---: |
> | OrthoHash | 53.94 | 16.01 | 7.27 | 11.05 | 56.43 | 15.88 | 7.49 | 11.15 | 61.72 | 14.54 | 7.10 | 10.64 | 60.53 | 17.05 | 8.46 | 11.85 |
> | Bihalf | 54.10 | 21.35 | 8.05 | 15.20 | 55.20 | 20.40 | 7.45 | 14.70 | 56.05 | 19.80 | 7.10 | 14.20 | 55.30 | 18.90 | 6.95 | 13.30 |
> | CE | 59.09 | 17.10 | 8.81 | 11.78 | 60.06 | 16.84 | 7.44 | 12.08 | 59.85 | 17.89 | 8.27 | 12.53 | 60.49 | 17.83 | 8.40 | 12.32 |
> | CSQ | 59.27 | 16.96 | 7.06 | 12.89 | 59.46 | 16.85 | 7.14 | 12.74 | 58.36 | 16.26 | 7.93 | 10.78 | 56.07 | 16.51 | 6.44 | 13.06 |
> | DFH | 58.55 | 13.36 | 5.45 | 10.91 | 61.03 | 15.42 | 6.39 | 12.04 | 59.94 | 14.91 | 5.88 | 12.10 | 57.80 | 15.60 | 6.54 | 11.93 |
> | DPSH | 57.59 | 15.63 | 6.72 | 12.57 | 57.25 | 18.42 | 8.83 | 12.85 | 58.30 | 17.63 | 8.18 | 12.92 | 59.68 | 18.25 | 8.41 | 12.88 |
> | DTSH | 57.27 | 15.57 | 6.53 | 12.52 | 61.29 | 15.32 | 6.26 | 12.66 | 57.66 | 15.98 | 6.44 | 12.03 | 57.57 | 15.22 | 5.86 | 12.60 |
> | GreedyHash | 65.03 | 16.39 | 7.44 | 12.68 | 60.39 | 15.71 | 7.13 | 11.82 | 63.58 | 18.89 | 9.26 | 12.86 | 60.33 | 17.30 | 7.58 | 12.65 |
> | SDH-C | 62.72 | 16.99 | 7.95 | 12.05 | 62.26 | 18.42 | 8.53 | 12.75 | 62.41 | 19.29 | 9.51 | 12.93 | 61.93 | 18.18 | 8.80 | 12.05 |
> | DLBD | 37.20 | 22.10 | 11.85 | 11.25 | 39.60 | 22.45 | 12.10 | 11.60 | 40.10 | 23.30 | 13.40 | 10.50 | 41.20 | 24.05 | 13.10 | 11.90 |
> | MDSHC | 64.80 | 14.20 | 5.60 | 9.25 | 62.10 | 13.40 | 5.95 | 9.48 | 61.70 | 12.30 | 5.40 | 10.95 | 61.20 | 12.55 | 5.20 | 10.60 |
> | FATE | 69.80 | 8.67 | 4.26 | 8.38 | 70.99 | 8.81 | 4.02 | 8.98 | 65.32 | 9.91 | 3.64 | 10.08 | 67.32 | 11.11 | 4.89 | 9.71 |
> | **DISH** | **70.13** | **7.41** | **3.32** | **7.91** | **71.48** | **7.87** | **2.98** | **8.86** | **71.10** | **8.77** | **3.01** | **9.52** | **72.06** | **7.67** | **2.79** | **8.60** |
>
> Together, these results demonstrate that DISH is not restricted to simple, binary sensitive attributes: it retains strong performance and fairness even in **multi-class** and **multi-dimensional (intersectional)** sensitive-attribute scenarios.

---

> ### Author Response · Authors · 2025-11-26
>
> **W3: Concern about computational cost and model complexity**
>
> **R3:** Thank you for identifying this critical issue. To quantify this, we measured the **per-epoch training time** of DISH and FATE on UTKFace under identical hardware and optimization settings, varying both the **backbone network** and **batch size**.
>
>
> ### Table 4. Per-epoch training time (UTKFace)
>
> | Method | Batch size | Backbone  | Time / epoch (s) |
> | :----- | :--------: | :-------- | :--------------: |
> | FATE   | 32         | ResNet-50 | 100.38           |
> | FATE   | 64         | ResNet-50 | 96.49            |
> | FATE   | 32         | AlexNet   | 24.36            |
> | FATE   | 64         | AlexNet   | 17.61            |
> | DISH   | 32         | ResNet-50 | 127.70           |
> | DISH   | 64         | ResNet-50 | 123.27           |
> | DISH   | 32         | AlexNet   | 22.22            |
> | DISH   | 64         | AlexNet   | 20.55            |
>
> Our results (Table 4) show that the **dominant factor in training cost is the choice of backbone**, while the additional modules in DISH (channel-wise heads, adversarial branch, covariance regularizer) are relatively lightweight compared to the backbone and do not fundamentally limit scalability.

---

> ### Author Response · Authors · 2025-11-26
>
> **Q1: Clarification on the specific semantics of the "factors" learned by the unentangled encoder.**
>
> **R4:** Thank you for raising this important point. We clarify that the "factors" in DISH are not explicitly pre-defined human-interpretable attributes (such as specific object parts, colors, or textures), but are **latent discriminative components** learned in a purely data-driven manner.
>
> Specifically, these factors are shaped through our **Disentangled Consistency Learning (DCL)** mechanism. By leveraging $K$ learnable prototypes combined with a supervised contrastive objective, the model automatically discovers and concentrates task-relevant semantic evidence into distinct, approximately independent factor-specific channels. This process ensures that the learned factors align with the underlying latent variations that maximize label consistency and class separability, rather than being manually assigned specific meanings. This factor-level decomposition is crucial as it allows the subsequent masking and adversarial modules to isolate and suppress sensitive information effectively.
>
>
> **Q2 Interactions and redundancy among components.**
>
> **R5:** Thank you for highlighting this issue. We clarify that the three core components—**Disentangled Consistency Learning (DCL)**, **Channel Adversarial Learning (CAL)**, and **Conditional Covariance Regularization (CCR)**—are designed to target theoretically distinct aspects of the representation learning process. Our analysis confirms that they are complementary rather than redundant, exhibiting strong synergistic effects.
>
> **1. Theoretical Distinctness**
> The components operate at different levels of the latent structure, ensuring no functional overlap:
> * **DCL (Structure Construction):** DCL is the foundational module. It minimizes the evidence lower bound to construct the factor-specific semantic structure and compute the assignment probabilities $p_\theta(k|\mathbf{x})$. Without DCL, the latent space lacks the disentangled organization required for the subsequent fairness modules to operate on specific factors.
> * **CAL (Intra-Channel Purification):** CAL acts **locally**. It minimizes the mutual information $I(S; \tilde{\mathbf{X}}_k)$ within each isolated channel, ensuring that individual factors do not explicitly encode sensitive attributes.
> * **CCR (Inter-Channel Decorrelation):** CCR acts **globally**. Even if individual channels are fair, sensitive information can be recovered through correlations *between* channels. CCR explicitly penalizes conditional covariance to prevent such "complementary coding," a leakage source that CAL cannot detect.
>
> **2. Empirical Analysis of Interactions**
> The ablation results in Table 4 reveal a mutually reinforcing relationship rather than diminishing returns:
> * **Synergy between Structure and Fairness:** While DCL alone yields only marginal gains (MAP ~62.14%), combining it with fairness objectives (e.g., DCL + CAL) triggers a substantial jump in accuracy to 68.07%. This suggests that adversarial debiasing is significantly more effective when applied to the semantically structured, factor-aligned space provided by DCL.
> * **Synergy between CAL and CCR:** The interaction between the two fairness objectives is critical. Adding CCR to the "DCL + CAL" baseline boosts MAP by over **5%** (68.07% $\rightarrow$ 73.33%) while nearly **halving** the Equalized Odds Difference (7.23% $\rightarrow$ 3.66%). This indicates that eliminating cross-channel correlations (CCR) simplifies the optimization landscape for the adversarial discriminators (CAL), allowing the model to achieve a Pareto frontier that neither component could reach in isolation.
>
>
> In light of these responses, we hope we have addressed your concerns, and we hope you will consider raising your score. We will properly include all the rebuttal contents in the revised version, following your valuable suggestions.

---

### Official Review · Reviewer_bMuF · 2025-11-01

**Soundness:** 3
**Presentation:** 4
**Contribution:** 2
**Rating:** 4
**Confidence:** 4

**Summary:**

This paper addresses fairness issues in deep hashing–based image retrieval systems, which can produce biased results across demographic groups. The authors propose DISH, a fairness-aware framework that learns disentangled and debiased hash representations in the continuous feature space before binarization. The model introduces three key modules: Disentangled Consistency Learning, which stabilizes factor-level semantics under augmentations and enforces interpretability; Information Suppressed Learning, which mitigates sensitive information leakage using probability-driven channel masking, channel-wise adversarial learning, and conditional covariance regularization; and Semantic Alignment, which maintains discriminative power in the Hamming space. Theoretically, the framework minimizes mutual information between sensitive attributes and channel representations, providing formal fairness guarantees. Empirical results on UTKFace and CelebA datasets demonstrate that DISH achieves state-of-the-art retrieval accuracy and fairness, outperforming prior methods such as FATE. Ablation studies further validate that each component contributes to improved fairness–utility trade-offs.

**Strengths:**

* The paper clearly identifies the under-explored issue of fairness in deep hashing–based image retrieval and motivates the need for addressing fairness before the binarization stage. The authors effectively argue that operating in the continuous feature space allows flexible and effective bias suppression.
* Extensive experiments on multiple benchmarks (UTKFace and CelebA) demonstrate clear Pareto improvements — DISH achieves higher retrieval accuracy and better fairness metrics (DP, EOP, EOD) than several baselines. The consistency across different settings adds credibility to the method’s robustness.
* Various ablation and component analysis. The paper includes detailed ablations and comparisons (e.g., different masking strategies, loss components, adversarial granularity) that clarify the contribution of each module. This level of experimental transparency is commendable and strengthens the empirical claims.

**Weaknesses:**

* Limited methodological novelty. The overall framework of DISH largely follows the established fair representation learning paradigm—combining disentangled representation learning with adversarial debiasing to suppress sensitive information. Similar concepts have been explored in prior works such as Adversarial Fair Representation (Zhang et al., 2018), β-VAE (Higgins et al., 2017), FactorVAE (Kim & Mnih, 2018), FFVAE (Creager, 2019) and fairness-oriented disentanglement models like Zhu et al. (2024) and Zhang et al. (2025) that explicitly separate semantic and sensitive attributes using adversarial objectives. Compared with these, the main difference of DISH is the addition of a binary hashing loss for retrieval; thus, the core contribution feels incremental rather than conceptually novel.
* Insufficient comparison to existing fair-representation methods. The paper does not thoroughly analyze how DISH compares to or improves upon well-known fair-representation frameworks such as Adversarial Fair Representation (Zhang et al., 2018) or Fair-VAE-style models. It remains unclear why adding the hashing loss directly to these existing methods would not achieve similar fairness-accuracy trade-offs. More empirical or theoretical justification for integrating disentanglement with binary hashing would strengthen the claim of methodological superiority.
* Limited exploration of training pipelines. The work assumes that fairness should be enforced jointly during hashing training, but does not evaluate a two-stage approach—e.g., learning a fair representation first and then fine-tuning with a hashing objective. Such baselines would clarify whether joint optimization is indeed necessary or simply convenient.
* High sensitivity to hyperparameters and training cost. The framework introduces at least two major hyperparameters (λ₁ and λ₂) to balance adversarial and covariance regularization terms, in addition to several architectural choices such as the number of channels K. The paper does not provide a principled method for setting these values, raising concerns about scalability, reproducibility, and tuning cost for deployment on larger or more complex datasets.

**Questions:**

* The proposed DISH framework integrates disentanglement, adversarial debiasing, and hashing into a single pipeline. However, several existing fair-representation methods (e.g., Adversarial Fair Representation, FairVAE, or disentanglement-based adversarial models like Zhu et al., 2024; Zhang et al., 2025) could potentially be combined with a binary hashing loss in a similar way. Could the authors clarify why  disentanglement, adversarial debiasing are better than these baselines in terms of combination of the hashing loss?

---

> ### Author Response · Authors · 2025-11-26
>
> **For Reviewer bMuF**
> We thank the reviewer for the constructive and detailed feedback. Below, we provide point-by-point responses and new experimental evidence to clarify our contributions.
>
> ---
> **W1: Concerns regarding methodological novelty compared to existing fair representation paradigms.**
>
> **R1:** Thanks for your valuable comment! While DISH adheres to the general learn a fair representation + adversarial debiasing paradigm, our contribution extends beyond simply attaching a hashing loss to an existing fair encoder. The core innovation lies in how we structure the latent space to be explicitly compatible with fair binary retrieval. We clarify the methodological distinctions below:
>
> * **Factor-Structured vs. Single Latent Vectors:**
>     Existing methods (e.g., $\beta$-VAE, AFR, FactorVAE) typically learn a single global latent vector. In contrast, DISH decomposes features into $K$ factor-specific channels using a supervised contrastive objective (DCL) with a variational posterior. This design concentrates semantic evidence into a *subset* of channels, allowing precise control over which channels propagate into the hash codes—a granular control mechanism absent in prior single-vector approaches.
>
> * **Structured Channel-wise Fairness:**
>     Unlike prior methods that rely on a single global adversary, DISH introduces a structured, three-part fairness mechanism designed for the discrete space:
>     1.  **Probability-driven Masking:** dynamically filters information based on learned factor assignment.
>     2.  **Channel-wise Adversarial Debiasing:** employs parallel discriminators per factor channel.
>     3.  **Conditional Covariance Regularization:** explicitly suppresses the re-emergence of sensitive cues through cross-channel interactions.
>
> * **Information-Theoretic Guarantees:**
>     We provide a theoretical analysis showing that reducing channel-wise mutual information $I(S; \tilde{X}_k)$ yields an explicit upper bound on the fairness gaps of the downstream hash codes:
>     $$
>     I(S; B) \le \sum\_{k} I(S; \tilde{X}\_k)
>     $$
> In summary, DISH is not a trivial combination of existing components but a **retrieval-specific architecture** that enforces fairness at the *factor level* to ensure robustness during quantization.We have **revised the Introduction in the updated manuscript** to explicitly discuss these methodological distinctions and better position DISH within the broader literature.

---

> ### Author Response · Authors · 2025-11-26
>
> **W2: Request for empirical comparison against fair representation baselines combined with a hashing loss.**
>
>
> **R2:** Thank you for this insightful comment. We have conducted new **end-to-end joint-training experiments** where we attached the same hashing head and semantic alignment loss to some fair-representation baselines (AFR, $\beta$-VAE, FactorVAE, FFVAE, SAD, CDFG) and trained them under settings identical to DISH.
>
>
>
> ### Table 1. UTKFace (Target: age, Sensitive: ethnicity) — Joint end-to-end training results:
>
> | Method    | MAP@16 | EOD@16 | EOP@16 | DP@16 | MAP@32 | EOD@32 | EOP@32 | DP@32 | MAP@64 | EOD@64 | EOP@64 | DP@64 | MAP@128 | EOD@128 | EOP@128 | DP@128 |
> |-----------|--------|--------|--------|-------|--------|--------|--------|-------|--------|--------|--------|-------|---------|---------|---------|--------|
> | β-VAE     | 28.20  | 11.36  | 6.56   | 5.45  | 31.69  | 14.53  | 8.30   | 7.29  | 30.52  | 14.83  | 8.04   | 7.35  | 30.92   | 14.61   | 8.22    | 6.85   |
> | FactorVAE | 31.68  | 8.93   | 5.03   | 5.12  | 25.12  | 9.12   | 3.49   | 7.85  | 25.56  | 8.64   | 4.22   | 6.31  | 27.16   | 10.08   | 5.50    | 6.34   |
> | FFVAE     | 25.13  | 9.18   | 4.66   | 9.92  | 24.68  | 9.49   | 4.49   | 10.38 | 24.62  | 9.26   | 4.44   | 10.21 | 25.52   | 9.02    | 4.53    | 9.88   |
> | AFR       | 63.33  | 9.82   | 6.44   | 6.69  | 67.29  | 10.50  | 6.65   | 7.33  | 62.22  | 9.02   | 5.23   | 6.27  | 66.31   | 13.20   | 8.94    | 7.62   |
> | SAD       | 55.58  | 8.35   | 4.65   | 6.55  | 61.97  | 6.57   | 3.36   | 7.07  | 63.21  | 7.47   | 4.38   | 5.93  | 58.70   | 8.98    | 4.72    | 6.99   |
> | CDFG      | 56.94  | 6.53   | 3.58   | 5.23  | 57.33  | 10.14  | 5.29   | 7.85  | 65.86  | 6.85   | 3.97   | 6.22  | 62.43   | 8.65    | 5.22    | 6.22   |
> | **DISH**  | **72.99** | **4.22** | **2.08** | **4.79** | **73.20** | **4.12** | **2.55** | **6.54** | **73.67** | **4.28** | **2.96** | **5.63** | **73.33** | **3.66** | **2.37** | **6.10** |
>
>
>
>  ### Table 2. UTKFace (Target: ethnicity, Sensitive: age) — Joint end-to-end training results:
>
> | Method    | MAP@16 | EOD@16 | EOP@16 | DP@16 | MAP@32 | EOD@32 | EOP@32 | DP@32 | MAP@64 | EOD@64 | EOP@64 | DP@64 | MAP@128 | EOD@128 | EOP@128 | DP@128 |
> |-----------|--------|--------|--------|-------|--------|--------|--------|-------|--------|--------|--------|-------|---------|---------|---------|--------|
> | β-VAE     | 31.42  | 11.53  | 6.71   | 10.38 | 31.59  | 21.31  | 11.05  | 11.02 | 32.51  | 15.15  | 9.21   | 9.83  | 31.34   | 20.47   | 11.12   | 9.99   |
> | FactorVAE | 24.07  | 16.69  | 6.81   | 10.81 | 21.74  | 13.54  | 4.48   | 10.48 | 21.68  | 9.45   | 3.59   | 9.79  | 24.34   | 19.03   | 9.82    | 9.77   |
> | FFVAE     | 23.12  | 14.85  | 5.78   | 9.92  | 22.05  | 12.61  | 4.02   | 9.88  | 21.94  | 9.12   | 3.01   | 9.24  | 22.86   | 17.25   | 8.96    | 9.11   |
> | AFR       | 57.67  | 12.28  | 6.78   | 8.21  | 59.05  | 12.38  | 6.22   | 9.33  | 57.21  | 14.33  | 6.92   | 10.38 | 60.64   | 14.94   | 6.86    | 11.69  |
> | SAD       | 51.11  | 12.99  | 5.59   | 9.81  | 58.46  | 12.79  | 4.91   | 11.22 | 56.26  | 13.85  | 6.43   | 10.71 | 56.99   | 10.37   | 3.89    | 9.23   |
> | CDFG      | 51.91  | 12.68  | 5.50   | 9.49  | 53.41  | 12.13  | 4.22   | 10.80 | 55.88  | 11.89  | 4.67   | 9.85  | 57.93   | 10.47   | 4.55    | 8.76   |
> | **DISH**  | **72.08** | **6.46** | **1.41** | **7.08** | **71.85** | **6.76** | **1.59** | **8.14** | **71.59** | **6.03** | **1.29** | **8.83** | **72.40** | **6.34** | **2.09** | **8.51** |
>
> The results (please refer to **Tables 8 and 9** of the revision for more detail ) demonstrate that simply adding a hashing loss to these methods fails to produce effective fair hash codes:
> * **VAE-based methods** ($\beta$-VAE, FactorVAE, FFVAE) yield poor retrieval performance (MAP $\approx$ 20–31%) and large fairness gaps, as their reconstruction-based latent spaces are not discriminative enough for hashing.
> * **Fairness-oriented disentanglement baselines** (AFR, SAD, CDFG) perform better but consistently fall short of DISH in both MAP and fairness metrics (EOD/EOP/DP).
>
> These results confirm that naively integrating a hashing objective into existing fair representation frameworks is insufficient to achieve an optimal fairness-accuracy trade-off.

---

> ### Author Response · Authors · 2025-11-26
>
> **W3: Suggestion to evaluate a two-stage training baseline (fair representation learning followed by hashing).**
>
> **R3:** Thank you for your valuable comment. Following your suggestion, we implemented a **two-stage training scheme**: (Stage 1) learn a fair continuous representation without hashing; (Stage 2) freeze the representation and train a hashing head.
>
> The results (please refer to **Tables 10 and 11** of the revision for more detail ) indicate that decoupling fairness learning from hashing significantly degrades performance.
> * **For DISH:** The two-stage variant yields strictly lower MAP and worse fairness metrics compared to our joint optimization.
> * **For Baselines:** The two-stage approach further inflates fairness gaps compared to the joint baseline experiments.
>
>
>
>  ### Table 3. Two-stage pipeline's results on UTKFace (Target: ethnicity, Sensitive: age)
>
> | Method     | MAP@16 | EOD@16 | EOP@16 | DP@16 | MAP@32 | EOD@32 | EOP@32 | DP@32 | MAP@64 | EOD@64 | EOP@64 | DP@64 | MAP@128 | EOD@128 | EOP@128 | DP@128 |
> |------------|--------|--------|--------|-------|--------|--------|--------|-------|--------|--------|--------|-------|---------|---------|---------|--------|
> | β-VAE      | 29.18  | 15.47  | 9.54   | 7.63  | 32.14  | 18.72  | 11.48  | 9.57  | 31.03  | 18.82  | 11.05  | 9.36  | 31.31   | 18.64   | 11.43   | 9.05   |
> | FactorVAE  | 25.63  | 12.58  | 7.46   | 7.85  | 24.74  | 13.06  | 5.25   | 9.96  | 25.33  | 12.04  | 6.16   | 8.47  | 27.71   | 14.73   | 7.95    | 8.67   |
> | FFVAE      | 25.47  | 13.25  | 7.66   | 12.57 | 25.16  | 13.46  | 7.18   | 12.86 | 24.98  | 12.84  | 6.86   | 12.38 | 25.82   | 13.06   | 7.32    | 12.05  |
> | AFR        | 63.82  | 13.27  | 8.95   | 9.36  | 67.72  | 15.03  | 9.47   | 10.14 | 62.81  | 13.16  | 7.95   | 9.04  | 66.92   | 17.33   | 11.58   | 10.47  |
> | SAD        | 56.04  | 11.46  | 6.63   | 8.74  | 62.31  | 10.17  | 5.47   | 9.33  | 63.92  | 10.74  | 6.27   | 8.05  | 59.14   | 11.82   | 6.43    | 9.16   |
> | CDFG       | 57.46  | 10.27  | 5.94   | 7.65  | 57.84  | 13.72  | 7.33   | 10.25 | 66.22  | 10.44  | 6.26   | 8.53  | 62.97   | 12.47   | 7.53    | 8.65   |
> | **DISH**   | **70.71** | 13.69 | **5.58** | **6.31** | **70.95** | 10.65 | 6.72 | **8.35** | **73.44** | **9.04** | **5.78** | **7.35** | **71.99** | **10.45** | **6.23** | 8.98 |
>
>
>
>  ### Table 4.Two-stage pipeline's results on UTKFace(Target: age, Sensitive: ethnicity)
>
> | Method     | MAP@16 | EOD@16 | EOP@16 | DP@16 | MAP@32 | EOD@32 | EOP@32 | DP@32 | MAP@64 | EOD@64 | EOP@64 | DP@64 | MAP@128 | EOD@128 | EOP@128 | DP@128 |
> |------------|--------|--------|--------|-------|--------|--------|--------|-------|--------|--------|--------|-------|---------|---------|---------|--------|
> | β-VAE      | 32.02  | 13.53  | 7.71   | 11.18 | 34.39  | 22.81  | 11.85  | 12.02 | 33.81  | 16.15  | 9.91   | 10.73 | 34.94   | 22.17   | 12.32   | 11.09  |
> | FactorVAE  | 24.67  | 18.69  | 7.81   | 11.61 | 22.54  | 15.04  | 5.28   | 11.48 | 21.98  | 10.45  | 4.29   | 10.69 | 21.24   | 20.73   | 11.02   | 10.87  |
> | FFVAE      | 21.72  | 16.85  | 6.78   | 10.72 | 19.85  | 14.11  | 4.82   | 10.88 | 21.24  | 13.52  | 5.76   | 11.26 | 20.46   | 18.95   | 10.16   | 10.21  |
> | AFR        | 58.27  | 14.28  | 7.78   | 9.01  | 59.85  | 13.88  | 7.02   | 10.33 | 57.51  | 15.33  | 7.62   | 11.28 | 60.24   | 16.64   | 8.06    | 12.79  |
> | SAD        | 51.71  | 14.99  | 6.59   | 10.61 | 59.26  | 14.29  | 5.71   | 12.22 | 56.56  | 14.85  | 7.13   | 11.61 | 56.59   | 12.07   | 5.09    | 10.33  |
> | CDFG       | 49.83  | 14.68  | 6.50   | 10.29 | 54.21  | 13.63  | 5.02   | 11.80 | 56.18  | 12.89  | 5.37   | 10.75 | 50.25   | 14.00   | 6.35    | 10.86  |
> | **DISH**   | **65.72** | **10.07** | **4.01** | **8.67** | **64.50** | 13.57 | 6.37 | **10.22** | **66.43** | **10.12** | **3.71** | **10.14** | **68.33** | **10.38** | **3.89** | **9.92** |
>
> These results confirm that fairness constraints must be enforced jointlywith the hashing objective to shape a latent space that remains robust to binarization.

---

> ### Author Response · Authors · 2025-11-26
>
> **W4: Concerns regarding hyperparameter sensitivity ($\lambda_1, \lambda_2$) and scalability.**
>
> **R4:** Thank you for raising this important point. We have extended our analysis to address concerns about tuning costs and generalizability:
>
> 1.  **Sensitivity Analysis:** We performed a comprehensive 2D sweep of $\lambda_1$ and $\lambda_2$. Figure 4 in the revision reveals a **stable ridge** where DISH consistently maintains high performance when $\lambda_1, \lambda_2 \in [0.01, 0.1]$. This stability allows for a simple initialization rule rather than exhaustive search.
> 2.  **Scalability & Generalization:** To validate the method's robustness on a larger scale, we evaluated DISH on **DeepFashion**, a challenging real-world dataset containing over 289,000 diverse images of clothing with complex variations in pose, zoom, and background. Crucially, we applied the **exact same hyperparameter configuration** derived from UTKFace to this new domain without any dataset-specific tuning. As shown below, DISH outperforms all state-of-the-art baselines.
>
>  ### Table 5.DeepFashion Results(Target Attribute: *attribute*,Sensitive Attribute: *category*)
>
> | Method | MAP@16 | EOD@16  | EOP@16  | DP@16  | MAP@32  | EOD@32  | EOP@32  | DP@32  | MAP@64  | EOD@64  | EOP@64  | DP@64  | MAP@128  | EOD@128  | EOP@128  | DP@128  |
> | :--- | :---: | :---: | :---: | :---: | :---: | :---: | :---: | :---: | :---: | :---: | :---: | :---: | :---: | :---: | :---: | :---: |
> | OrthoHash | 28.09 | 9.74 | 5.79 | 4.03 | 29.71 | 6.20 | 3.91 | 2.26 | 30.00 | 6.77 | 4.07 | 2.66 | 31.53 | 7.26 | 4.86 | 2.43 |
> | Bihalf | 33.52 | 25.11 | 14.98 | 10.80 | 30.08 | 23.10 | 13.14 | 10.53 | 27.35 | 17.65 | 10.64 | 7.48 | 26.97 | 22.80 | 13.22 | 10.05 |
> | CE | 28.53 | 8.96 | 4.98 | 4.05 | 29.21 | 7.56 | 4.44 | 3.09 | 29.18 | 9.10 | 5.26 | 3.86 | 30.05 | 9.46 | 5.40 | 4.13 |
> | CSQ | 30.42 | 7.08 | 4.21 | 3.01 | 32.33 | 17.09 | 10.20 | 7.21 | 33.48 | 13.50 | 8.42 | 5.37 | 32.44 | 17.04 | 9.82 | 7.43 |
> | DFH | 37.47 | 7.65 | 5.42 | 2.55 | 33.64 | 11.27 | 7.19 | 4.45 | 32.70 | 15.27 | 9.58 | 6.21 | 31.41 | 13.24 | 8.62 | 5.06 |
> | DPSH | 32.17 | 7.65 | 4.65 | 3.07 | 32.02 | 7.11 | 4.20 | 2.90 | 32.08 | 6.35 | 3.83 | 2.47 | 32.71 | 5.64 | 3.46 | 2.15 |
> | DTSH | 30.97 | 5.98 | 3.46 | 2.47 | 34.23 | 9.02 | 5.43 | 3.85 | 30.47 | 9.80 | 6.47 | 3.57 | 30.02 | 8.30 | 5.71 | 2.83 |
> | GreedyHash | 31.19 | 11.17 | 6.99 | 4.51 | 29.67 | 7.12 | 4.24 | 2.88 | 30.06 | 7.62 | 4.33 | 3.27 | 29.21 | 9.92 | 5.40 | 4.60 |
> | SDH-C | 29.83 | 11.30 | 6.82 | 4.55 | 29.43 | 13.76 | 8.14 | 5.78 | 29.34 | 14.65 | 8.63 | 6.38 | 27.77 | 17.49 | 9.91 | 7.93 |
> | DLBD | 23.88 | 14.67 | 6.67 | 7.81 | 24.14 | 13.47 | 6.21 | 7.16 | 24.38 | 12.90 | 5.88 | 6.88 | 24.51 | 13.73 | 6.25 | 7.37 |
> | MDSHC | 27.79 | 8.22 | 3.97 | 4.16 | 27.01 | 8.37 | 4.50 | 3.80 | 26.48 | 14.24 | 7.21 | 7.03 | 24.55 | 14.50 | 6.81 | 7.60 |
> | FATE | 41.46 | 3.90 | 2.70 | 2.50 | 42.61 | 4.60 | 2.40 | 2.10 | 42.89 | 3.93 | 1.96 | 1.90 | 42.35 | 3.50 | 1.90 | 1.70 |
> | **DISH** | **47.13** | **2.96** | **1.65** | **1.49** | **48.53** | **2.86** | **1.83** | **1.77** | **48.40** | **2.74** | **1.81** | **1.87** | **48.98** | **2.45** | **1.84** | **1.63** |
>
> These results collectively demonstrate that DISH is not excessively sensitive to hyperparameter choices and can be reliably deployed on large-scale, complex datasets without high tuning costs.

---

> ### Author Response · Authors · 2025-11-26
>
> ---
>
> **Q1: Clarification on why DISH’s integration of disentanglement and debiasing outperforms baselines with added hashing loss.**
>
>
> **R5:** Thank you for raising this important point. Our new experiments (Tables1–4) directly address this by comparing DISH against "Baseline + Hashing Loss" in both joint and two-stage settings. The empirical evidence shows that while baselines like AFR or FairVAE can learn fair *continuous* embeddings, they fail to produce representations that remain semantically discriminative and fair when binarized.
>
> The superiority of DISH stems from its architectural design:
> 1.  **Factor-Level Decomposition:** Unlike global vectors, our channel-wise approach isolates semantic vs. sensitive attributes *before* quantization.
> 2.  **Robustness to Quantization:** By enforcing fairness (via masking and covariance regularization) specifically on the factor channels during joint training, DISH ensures that the binarization process does not destroy the learned fairness-accuracy trade-off.
>
> In short, simply attaching a hashing loss to a fair continuous model is insufficient because continuous fairness does not guarantee binary fairness; DISH bridges this gap through its specialized factor-structured design.We have incorporated this analysis into the **Discussion** section of the revised manuscript, with full quantitative comparisons provided in the **Appendix F**.
>
> [1] Zhang, B. H., Lemoine, B., & Mitchell, M. Mitigating unwanted biases with adversarial learning. AIES 2018, pp. 335-340.
>
> [2] Higgins, I., Matthey, L., Pal, A., Burgess, C., Glorot, X., Botvinick, M., ... & Lerchner, A. beta-vae: Learning basic visual concepts with a constrained variational framework. ICLR 2017.
>
> [3] Kim, H., & Mnih, A. Disentangling by factorising. ICML 2018, pp. 2649-2658.
>
> [4] Zhu, Y., Li, J., Zheng, Z., & Chen, L. Fair graph representation learning via sensitive attribute disentanglement. WWW 2024, pp. 1182-1192.
>
> [5] Zhang, G., Cheng, D., Yuan, G., Liu, S., & Zhang, Y. Causality-inspired disentanglement for fair graph neural networks. IJCAI 2025, pp. 637-645.
>
> In light of these responses, we hope we have addressed your concerns, and we hope you will consider raising your score. We will properly include all the rebuttal contents in the revised version, following your valuable suggestions.

---

### Meta-Review · Area_Chair_kbjb · 2026-01-08

**Summary:**

Reviewers have raised the concerns about Limited methodological novelty, originality and significance. The approach combines disentangled representation learning with adversarial debiasing to suppress sensitive information. The authors have responded to this issue and addressed it to some extent, but the underlying contribution remains somewhat incremental.

Reviewers have also concerned the insufficient comparison to existing fair-representation methods, and inconsistent reported results from previous works. The authors have responded to this issue by complementing additional experiments. They also claimed that they have done rigorous reproduction using a ResNet-50 backbone, strictly adhering to previous method’s official training paradigm and settings.

A question is about the work is face-image only: this paper primarily focuses on the "face image retrieval" scenario. The authors have addressed this issue by adding an experiment on the DeepFashion dataset.

Overall, a complete rebuttal has been done. However, the method's novelty is moderate. In addition, the problem seems to be solvable at inference time. Eg., in DeepFashion, the sensitive attribute is the category. Since both the target and sensitive attributes are explicit in the gallery, a basic solution is to retrieve approximately equal numbers of samples for each category and then apply some post‑processing. The approach requires defining the target and sensitive attributes before training, which is not flexible enough for practical applications.

**Reviewer Concerns:**

The reviewers have raised concerns about method novelty, insufficient comparison and inconsistent reported results, the approach is for face-image only, restricted to single-dimensional sensitive attributes, using more modern backbone (e.g., Vision Transformer), and so on. The authors have provided a complete and thorough rebuttal.

**Reviewer Scores:**

I think most reviewers will maintain the original scores.

---

### Decision · Program_Chairs · 2026-01-26

Reject